



# The CAMS reanalysis of atmospheric composition

Antje Inness[1], Melanie Ades[1], Anna Agusti-Panareda[1], Jérôme Barré[1], Anna Benedictow[2], Anne-Marlene Blechschmidt[3], Juan Jose Dominguez[1], Richard Engelen[1], Henk Eskes[4], Johannes Flemming[1], Vincent Huijnen[4], Luke Jones[1], Zak Kipling[1], Sebastien Massart[1], Mark Parrington[1], Vincent-Henri Peuch[1], Miha Razinger[1], Samuel Remy[5], Michael Schulz[2] and Martin Suttie[1]

[1]ECMWF, Shinfield Park, Reading, RG2 9AX, UK
[2]Norwegian Meteorological Institute Postboks 43 Blindern, 0313 Oslo, Norway
[3]Institute of Environmental Physics, University of Bremen, Germany
[4]Royal Netherlands Meteorological Institute, De Bilt, The Netherlands
[5]IPSL, CNRS/UPMC, Paris, France

*Correspondence to*: Antje Inness (a.inness@ecmwf.int)

**Abstract.** The Copernicus Atmosphere Monitoring Service (CAMS) reanalysis is the latest global reanalysis data set of atmospheric composition produced by the European Centre for Medium-Range Weather Forecasts (ECMWF), consisting of 3-dimensional time-consistent atmospheric composition fields, including aerosols and chemical species. The dataset currently covers the period 2003-2016 and will be extended in the future by adding one year each year. A reanalysis for greenhouse gases is being produced separately. The CAMS reanalysis builds on the experience gained during the production of the earlier Monitoring Atmospheric Composition and Climate (MACC) reanalysis and CAMS interim reanalysis. Satellite retrievals of total column CO, tropospheric column $NO_2$, aerosol optical depth and total column, partial column and profile ozone retrievals were assimilated for the CAMS reanalysis with ECMWF's Integrated Forecasting System. The new reanalysis has an increased horizontal resolution of about 80 km and provides more chemical species at a better temporal resolution (3-hourly analysis fields, 3-hourly forecast fields and hourly surface forecast fields) than the previously produced CAMS interim reanalysis. The CAMS reanalysis has smaller biases compared to independent ozone, carbon monoxide, nitrogen dioxide and aerosol optical depth observations than the previous two reanalyses and is much improved and more consistent in time, especially compared to the MACC reanalysis. The CAMS reanalysis is a dataset that can be used to compute climatologies, study trends, evaluate models, benchmark other reanalyses or serve as boundary conditions for regional models for past periods.

## 1 Introduction

The European Centre for Medium Range Weather Forecasts (ECMWF) has been producing atmospheric composition (AC) forecasts and analyses for over a decade. The model and data assimilation system used for this was developed as a European effort by a consortium of partners funded by several European Union (EU) projects. It began in 2005, with the EU funded Global Monitoring for Environment and Security (GEMS) project (Hollingsworth et al. 2008) that built the capacity for a global and regional forecasting and data assimilation system of AC. In GEMS, ECMWF's Integrated Forecast System (IFS) was extended to allow for the data assimilation and modelling of aerosols, chemically reactive gases and greenhouse gases, and the first daily forecasts of reactive gases such as carbon monoxide (CO) and tropospheric ozone ($O_3$) were made public in May 2007 (Flemming et al., 2017a). This was followed a year later, in July 2008, by the real-time data assimilation of aerosol optical depth (Benedetti et al., 2009) and selected reactive gases (Inness et al., 2013) in the daily GEMS system. The AC system was further developed in the earlier Monitoring Atmospheric Composition and Climate (MACC) projects (Flemming et al., 2015; Inness et al., 2015; Massart et al., 2014; Agusti-Panareda et al., 2014) between 2009 and 2014 and has been running fully operationally in the Copernicus Atmosphere Monitoring Service (CAMS), operated by ECMWF, since January 2015. In the rest of this paper we will refer to the system built in GEMS/MACC/CAMS as the CAMS system and focus on reactive gases and aerosols.





While the modelling components for aerosols were included directly in the IFS from the beginning of GEMS (Morcrette et al., 2009), the initial approach for the reactive gases was to build a 'coupled system' where the chemical transport model (CTM) Model for OZone And Related chemical Tracers (MOZART) 3.5 (Kinnison et al., 2007) was coupled to the IFS using the

Ocean Atmosphere Sea Ice Soil (OASIS 4) coupling software (Flemming et al., 2009). Later, in the MACC projects, a modified version of the Carbon Bond 2005 chemistry scheme (CB05, Huijnen et al. 2010) derived from the CTM Transport Model 5 (TM5) was integrated in the IFS (referred to as IFS(CB05); Flemming et al., 2015), making the system more computationally efficient and improving the system's ability to represent interactions between meteorology and chemistry. In parallel to the model system, the use of observations in the data assimilation system also evolved with time as new datasets became available

and satellite retrievals were improved.

The initial forecasts and analyses in 2007/2008 had a horizontal resolution of T159 (~110 km). This was increased to T255 (~80 km) in July 2012 and to T511 (~40 km) in June 2016. The vertical resolution has so far always consisted of 60 model levels in the vertical, with the top level at 0.1 hPa. Details about the different model versions used over time are given in Table

A1 in the Appendix.

The continuing upgrades of the CAMS model and data assimilation system and the changes they brought with them made it difficult if not impossible to compare data from a recent period with earlier data in a meaningful way. For example, it was not possible to calculate seasonal anomalies or trends with a system that had changed so much over time. Therefore, so called

reanalyses were produced with the CAMS system where a long time period was re-run with a single version of the model and data assimilation system, taking care to minimise changes in the versions of the used emissions or assimilated satellite retrievals. Such a system gives the temporal consistency needed to deduce trends (e.g. Flemming et al., 2017b) or to provide maps of annual or seasonal anomalies (e.g. Flemming and Inness, 2018).

Producing long reanalyses with a single model version is a well-known procedure in Numerical Weather Prediction (NWP), and several weather centres have produced meteorological reanalysis data sets. It has long been an important activity at ECMWF (ERA-40, Uppala et al., 2005; ERA interim, Dee et al., 2011; ERA-5, Hersbach et al., in preparation), and other meteorological centres such as National Centers for Environmental Protection (NCEP) (CFSR; Saha et al., 2010), the Japan Meteorological Agency (JRA-55, Kobayashi et al., 2015;  JRA-25, Onogi et al., 2007), NASA-GMAO (Modern-Era

Retrospective analysis for Research and Applications (MERRA), Rienecker, et al., 2011; MERRA-2, Gelaro et al., 2017) and the  China Meteorological Administration (CRA-40) have also produced or are producing reanalyses.

In addition to these meteorological reanalyses several reanalyses of atmospheric composition have been produced in recent years. The multi-sensor reanalysis of total ozone (van der A et al., 2015) for 1970–2012 used ground-based Brewer

observations to inter-calibrate satellite retrievals. The MERRA reanalysis (1980 to 2016) included ozone and was used to drive an offline aerosol reanalysis (MERRAero; Buchard et al. 2014). The MERRA-2 (Gelaro et al., 2017) reanalysis, again from 1980 onwards, also contained aerosols (Randles et al., 2017; Buchard et al, 2017).  Miyazaki et al. (2015) put together a tropospheric chemistry reanalysis for the years 2005–2014 and the Meteorological Research Institute (MRI) of the Japan Meteorological Agency produced a 5-year aerosol reanalysis product (Japanese Reanalysis for Aerosol, JRAero) for the years

2011-2015 (Yumimoto et al., 2017).

ECMWF produced several AC reanalyses in the GEMS/MACC/CAMS projects (see Table A2 in the Appendix). All these reanalyses started in 2003, when a wealth of atmospheric composition retrievals became available after the launch of the



European Envisat satellite and the American Aqua and Aura satellites. The so-called 'GEMS reanalysis' was a 6-year reanalysis of reactive gases, aerosols and greenhouse gases covering the period from 2003 to April 2009. This was followed by the 10-year 'MACC reanalysis' for reactive gases and aerosols covering the period 2003 to 2012 (MACCRA, Inness et al., 2013). The GEMS and MACC reanalyses both used the coupled IFS/MOZART 3.5 system. After the change to the integrated
IFS(CB05) system in September 2014, the model as well as the data assimilation configuration changed considerably, and comparing fields from the later years with a climatology based on the MACC reanalysis showed mainly model and configuration differences and not a climatologically meaningful signal. Therefore, a new reanalysis run with IFS(CB05) was needed. To prepare for this, a test reanalysis for reactive gases and aerosols, the 'CAMS interim reanalysis' (CIRA, Flemming et al., 2017b), was produced with a version of the IFS(CB05) system from 2003 onwards. CIRA was run at lower horizontal
resolution (T159, ~110km) than the MACC reanalysis (T255, ~80 km) and the number of archived fields was slightly reduced to speed up the throughput. This helped to test aspects of the IFS(CB05) system and paved the way for the production of the 'CAMS reanalysis' (CAMSRA), again from 2003 onwards and with the IFS(CB05) system. CAMSRA includes reactive gases and aerosols at higher horizontal resolution (T255) and with an increased number and time frequency of archived fields. Further improvements of CAMSRA relative to CIRA are the assimilation of $NO_2$ retrievals in CAMSRA and a better representation
of the interannual variability of the biogenic emissions. A reanalysis for the greenhouse gases $CO_2$ and $CH_4$ is being produced separately and will be discussed in a different paper.

Figure 1a shows the 'Figure of Merit in space' (FMS; Chang and Hanna, 2004) ozone score at the Antarctic Neumayer Station and Figure 1b the modified normalised mean bias (MNMB) of CO in the lower troposphere from the CAMS daily forecast
system and CAMSRA to illustrate the advantage a reanalysis has over a continuously evolving operational model system. The definitions of the scores are given in the Appendix. The FMS score compares the fit of the model ozone profiles to ozonesonde profiles (here calculated from the surface to 3 hPa) and has a score between 1 (perfect fit) and 0. Figure 1 illustrates the improvements in the near real-time (NRT) CAMS system with time. In the earlier years the CAMS system did not adequately reproduce the low values and vertical distribution of the Antarctic ozone hole (Flemming et al., 2011; Inness et al., 2013)
which is shown by the low FMS scores in austral spring from 2008-2012 (Fig. 1a). The CAMSRA has much improved FMS scores during those years and a better consistency in performance between earlier and later years. The timeseries of MNMB of CO in the lower troposphere (Fig. 1b) also shows problems of the NRT system in the earlier years in the northern hemisphere (NH) during winter, when CO values were strongly underestimated (Inness et al., 2013). There is still an underestimation of lower tropospheric CO in the CAMSRA, but the MNMB is now considerably smaller, especially during NH winter, and more
constant in.

The aim of this paper is to document the new CAMSRA dataset for future reference. The paper gives information about the model and data assimilation setup used to produce CAMSRA, presents initial validation results and intercompares CAMSRA with CIRA and MACCRA. Additional validation of CAMSRA can be found in reanalysis validation reports (Eskes et al.,
2018) and in two validation papers that are in preparation (Wagner et al, in preparation; Yang et al., in preparation).

This paper is structured in the following way. Section 2 describes the CAMS model, the data assimilation system and the emission datasets used to produce the CAMS reanalysis. Section 3 lists the assimilated AC observations and bias correction procedure. Section 4 gives validation results for some of the reactive gases and aerosols, and Section 5 presents the conclusions.



## 2 CAMS model and data assimilation system

An overview of the main differences and commonalities of the three reanalyses MACCRA, CIRA and CAMSRA discussed in this paper is given in Table 1.

### 2.1 CAMS model system

The IFS aerosol and chemistry modules applied in CAMSRA were similar to those in CIRA and more details about the modules are given in Flemming et al. (2015) and references therein. Major updates relative to CIRA are described in the sections 2.1.1 and 2.1.2 below. The meteorological modelling part of the IFS changed from cycle 40R1 used for CIRA to cycle 42R1 used for CAMSRA (see also Table 1).

### 2.1.1 Aerosol model updates

The CAMS aerosol model component of the IFS was previously described in Morcrette et al. (2009). It is a hybrid bulk/bin scheme with 12 prognostic tracers, consisting of three bins for sea salt depending on size (0.03-0.5, 0.5-5 and 5-20$\mu$m), three bins for dust (0.030-0.55, 0.55-0.9 and 0.9-20 $\mu$m), hydrophilic and hydrophobic organic matter (OM) and black carbon (BC), plus sulphate aerosol and a gas-phase sulphur dioxide ($SO_2$) precursor. The different aerosol types are treated as externally mixed, i.e. separate particles. Transport by advection, convection and diffusion is handled by the meteorological model

component of the IFS. The aerosol scheme includes prescribed and online emissions (as described in Section 2.2), dry and wet deposition, production of sulphate from $SO_2$, and ageing of hydrophobic OM and BC to hydrophilic.

The aerosol model used in the CAMS reanalysis contains these updates relative to CIRA:
- Updated aerosol optical properties, especially for organic matter (as described in Bozzo et al., 2017)
- Bug fixes to sedimentation, which was unreasonably weak for some dust and sea-salt bins, with corresponding re-tuning of sea-salt scavenging
- $SO_2$ dry deposition velocities updated to use monthly values computed by Météo-France's Surface Module of the MOCAGE model (SUMO, Michou et al., 2004). They now match those used in the chemistry scheme.
- New parameterisation of anthropogenic Secondary Organic Aerosol (SOA) production, proportional to MACCity
CO emissions, as suggested in Spracklen et al. (2011)
- More detailed $SO_2$ to sulphate aerosol conversion with dependence on temperature and relative humidity, and overall decrease in the conversion timescale especially at high latitudes
- Increased sulphate dry deposition velocity over ocean
- Proportional mass fixer used for chemistry (Diamantakis and Flemming, 2014) extended to aerosol species
- In CIRA, emissions from the Global Fire Assimilation System (GFAS) of black carbon (BC) were scaled by a globally constant factor of 3.4, which had been derived by comparing BC from a 6-month assimilation run with a forecast only run. In CAMSRA the same approach was used but comparing 12 years of CIRA data against a control run without data assimilation. This made it possible to derive a geographically varying (but temporally constant) scaling factor for BC GFAS emissions in CAMSRA.
- The $SO_2$ emissions in CAMSRA are separated between emissions at low level (20% of total emissions, which are emitted as part of the diffusion scheme) and high level emissions (80% of total emissions which are released in the two lowest model levels). In CIRA all $SO_2$ emissions were released at the surface as part of the diffusion scheme.





### 2.1.2 Chemistry module updates

The chemical mechanism of the IFS is a modified and extended version of the CB05 (Yaarmond et al. 2005) chemical mechanism for the troposphere, as implemented in CTM TM5 (Huijnen et al., 2010). CB05 describes tropospheric chemistry with 55 species and 126 reactions. Stratospheric ozone chemistry in IFS(CB05) is parameterized by a "Cariolle-scheme"

(Cariolle and Déqué, 1986; Cariolle and Teyssèdre, 2007). Wet deposition is modelled following Jacob (2000), and monthly-mean gridded dry deposition velocities calculated by the SUMO model of Météo-France (Michou et al. 2004) were used to calculate dry deposition. The chemistry module of the IFS is documented in more detail in Flemming et al. (2015) and Flemming et al. (2017b). The following updates of the chemistry scheme from the configuration used in CIRA were applied to produce CAMSRA:

- Update of heterogeneous rate coefficients for $N_2O_5$ and $HO_2$ based on prognostic clouds and aerosol
    - Modification of photolysis rates by prognostic aerosol
    - Dynamic tropopause definition based on the temperature profile for coupling to the Cariolle scheme in the stratosphere and for tropospheric mass diagnostics
    - Bugfixes, in particular for the diurnal cycle of dry deposition whose correction has decreased the ozone dry
deposition flux by about 15-20%

It should be noted that the schemes for aerosol and chemistry in IFS(CB05) are largely independent, which means in particular that both the aerosol and the chemistry scheme carry their own $SO_2$ variable. The conversion of the aerosol $SO_2$ to sulphate aerosol is modelled in the aerosol scheme by prescribed conversion rates (Morcrette et al., 2009), whereas $SO_2$ in the chemistry

scheme is subject to gas-phase and aqueous phase chemistry. The sulphate of the chemistry scheme does not contribute to the aerosol optical properties nor is it corrected by data assimilation. However, the first steps to link chemistry and aerosol schemes have been undertaken and the aerosol model affects the chemical composition by using the aerosol surface area density in the heterogenous reaction rates of dinitrogen pentoxide ($N_2O_5$) and hydroperoxyl ($HO_2$) (Huijnen et al., 2014) as well as by using aerosol optical properties for the modification of photolysis rates.

A major difference between the production of CIRA and CAMSRA is that the prognostic ozone and aerosol fields have been used interactively in the NWP radiation scheme in CAMSRA. For CIRA climatologies of ozone derived from MACCRA (Bozzo et al., 2017) and the Tegen et al. (1997) aerosol climatology were used in the radiation scheme. The evaluation of the meteorological parameters is beyond the scope of this paper. Nevertheless, little differences were found by introducing

prognostic ozone and aerosol because the meteorological analysis is well constrained by the assimilated observations. Furthermore, no change in the evaluation of the AC parameters could be identified when using ozone and aerosol interactively in the radiation scheme.

### 2.2 Emissions

Great care has been taken to ensure that the emission datasets are consistent in time and that consistent anthropogenic, biogenic

and biomass burning emissions were used for the aerosol and chemistry fields. The emission datasets are listed in Table 1. The emissions are injected at the surface and distributed over the boundary layer by the model's convection and vertical diffusion scheme. The only exception is the aerosol $SO_2$ emissions of which 20% are emitted at the surface as part of the diffusion scheme and 80% in the two lowest model levels (see Section 2.1.1). The emissions datasets used in CAMSRA include emissions from anthropogenic, biogenic, natural, and biomass burning sources.

Anthropogenic emissions were from the MACCity inventory (Granier et al., 2011) with modifications to increase winter-time road traffic emissions over North America and Europe following the correction of Stein et al. (2014). These emissions were



also used in CIRA while they were used without the Stein et al. (2014) correction in MACCRA. The MACCity inventory covers the period 1960-2010 and is updated for subsequent years using the representative concentration pathway (RCP) version 8.5. The RCP 8.5 (business as usual) scenario was chosen as it includes information on regional emissions after 2000 (Van Vuuren et al., 2010; Riahi et al., 2011). The anthropogenic MACCity emissions for CO are shown in Fig. 2. They decrease

over Europe and North America in the range of 1 to 5 % per year, but increase over South East Asia by a similar amount. The global trend for CO is close to zero. Anthropogenic emissions of black carbon, organic carbon and $SO_2$ were also taken from MACCity. Emissions of anthropogenic SOA were estimated by applying a scaling factor of 0.2 to the MACCity (i.e., non-biomass burning) CO emissions, as suggested in Spracklen et al. (2011).

Monthly mean biogenic emissions of the chemical species were calculated offline by the MEGAN2.1 model (Guenther et al., 2006) that used meteorological fields from the MERRA-2 reanalysis following Sindelarova et al. (2014) for the full period of CAMSRA. Natural emissions from soils and oceans for $NO_2$, dimethyl sulphate (DMS) and $SO_2$ were taken from the Precursors of ozone and their effects in the Troposphere (POET) database for 2000 (Granier et al., 2005; Olivier et al., 2003).

Daily global biomass burning emissions of reactive gases and aerosols were provided by the Global Fire Assimilation System, version 1.2 (GFASv1.2), based on satellite retrievals of fire radiative power (Kaiser et al., 2012). The archive of GFASv1.2 data covers the period 2003-present and was also used in CIRA. In MACCRA early versions of the Global Fire Emissions Database (GFED 3.1 (van der Werf et al., 2010) were used from 2003 until the end of 2008 and daily GFAS v1.0 data from 2009 to 2012. GFED 3.1 is on average 20% lower than GFAS v1.2 (Inness et al., 2013). Figure 3 shows the GFASv1.2

timeseries of monthly mean total carbon wildfire emissions for each of the main continental regions, excluding Antarctica, between 2003 and 2016. The emissions from GFEDv3.1 and GFASv1.0 are also shown for comparison. The CO biomass burning emissions do not show a significant trend but considerable inter-annual variability. Africa is usually the largest source of CO biomass burning emissions, but under El Niño conditions Asian emissions (and in particular emissions from maritime Southeast Asia) reach similar values. Most notable here are the Asian emissions during the Indonesian fires in September and

October 2015 that caused by far the highest annual wildfire emissions as well as the highest total monthly CO emissions in the whole period covered by GFAS (Huijnen et al., 2016).

The aerosol model has additional online parameterisations to calculate sea salt (Monahan et al., 1986) and dust surface fluxes based on surface winds and other factors (Ginoux et al., 2001).

**2.3 CAMS data assimilation system**

The IFS uses an incremental 4D-Var data assimilation system (Courtier et al. 1994) for the CAMS analyses, with 12-hour assimilation windows from 09 UTC to 21 UTC and 21 UTC to 09 UTC and two minimisations at spectral truncations T95 (~210 km) and T159 (~110 km). In the CAMS 4D-Var a cost function that measures the differences between the model's background fields and the observations is minimized in order to obtain the best possible forecast through the length of the

assimilation window by adjusting the initial conditions. Several atmospheric composition fields (i.e. $O_3$, CO, $NO_2$ and total aerosol mass mixing ratio) are included in the control vector and minimized together with the meteorological control variables. The background errors for the atmospheric composition fields were either calculated with the National Meteorological Center (NMC) method (Parrish and Derber 1992) or from and ensemble of forecast differences (Inness et al., 2015). Both methods allow us to calculate differences between pairs of background fields which have the statistical characteristics of the background

errors. More information about the data assimilation system and background errors for the chemical fields can be found in Inness et al. (2015). The aerosol assimilation is described in Benedetti et al. (2009) and the background errors used for the aerosol assimilation in Benedetti and Fisher (2007).



The aerosol assimilation is less constrained than the assimilation of the chemical species because the model has 12 different aerosol components (see Section 2.1.1), while the assimilated observations are retrievals of total aerosol optical depth (AOD). Therefore, the total aerosol mass mixing ratio, defined as the sum of the aerosol species, is used as control variable and the

analysis increments are repartitioned into the individual aerosol components (the $SO_2$ precursor is excluded from this process, as it is not visible in the AOD observations) according to their fractional contribution to the total aerosol mass (Benedetti et al., 2009). Flemming et al. (2017b) have shown that this can lead to problems as the relative fraction of the aerosol components is not conserved during the whole assimilation procedure because of differences in aerosol lifetime associated with differences in their removal processes. Aerosol components with a longer atmospheric lifetime can retain the change imposed by the

increments for longer and may thereby change the relative contributions. Also, if the underlying aerosol model has a bias in one aerosol species, e.g. it overestimates the species and thereby contributes a bigger fraction to the total aerosol mass mixing ratio than it should do, the assimilation can exacerbate this by attributing a greater proportion of the increment to this species and enhancing the bias even further. This was the case in CIRA where it led to an unrealistic overestimation of sulphates (Flemming et al., 2017b).

**2.4 CAMSRA data product**

The spatial resolution of the CAMS reanalysis is a reduced gaussian grid at a spectral truncation of T255, which is equivalent to grid spacing of approximately 80 km globally (0.7º x 0.7° grid). The vertical model resolution consists of 60 hybrid sigma/pressure (model) levels with a model top at 0.1 hPa. The data are available as 3-hourly analyses and 48 hour forecasts, initialised daily from analyses at 00 UTC. Three-dimensional model level forecast fields are available every 3 hours from

forecast hour 0 to 48, and surface forecast fields are available at hourly intervals. Monthly mean fields are also provided. Atmospheric data are archived on 60 model levels and are also interpolated to 25 pressure, 10 potential temperature and 1 potential vorticity level(s). Surface and total column diagnostics are also available (https://software.ecmwf.int/wiki/display/CKB/CAMS+Reanalysis+data+documentation#CAMSReanalysisdatadocumentatio n-Parameterlistings). An inventory of the available model fields can be found here: http://apps.ecmwf.int/data-

catalogues/cams-reanalysis/?class=mc&expver=eac4, and more information at https://atmosphere.copernicus.eu/copernicus-releases-new-global-reanalysis-data-set-atmospheric-composition.

**3 Observations and bias correction**

**3.1 Observations**

The atmospheric composition satellite retrievals of $O_3$, CO, $NO_2$ and AOD that were assimilated to produce CAMSRA are

shown in Fig. 4 and listed in Table 2. The table also shows the blacklist criteria applied to the data, i.e. the criteria that determine when data were not used.

Retrievals from a range of instruments were used for $O_3$. These included total column $O_3$ (TCO3) retrievals from the SCanning Imaging Absorption spectroMeter for Atmospheric CHartographY (SCIAMACHY) instrument, the Ozone Monitoring

Instrument (OMI) and the Global Ozone Monitoring Experiment-2 (GOME-2), $O_3$ profile data from the Michelson Interferometer for Passive Atmospheric Sounding (MIPAS) and Microwave Limb Sounder (MLS, used down to 215 hPa) and $O_3$ partial columns from Solar Backscatter ULTra-Violet (SBUV/2). For SBUV/2 the v8.6 data record (McPeters et al. 2013) was used until July 2013 and NRT V8 data afterwards. The V8.6 data are available at 21 vertical layers but were converted into a 13-layer product by CAMS to reduce smoothing errors, by combining the layers between 16 hPa and the surface. The

NRT SBUV/2 data were the same data used in the CAMS real-time analysis and used at the 21L resolution. Changing to the





NRT data was necessary as the reprocessed data were not available after July 2013. We do not notice a change in the ozone analysis field at the time of this change because the analysis is well constrained by the other assimilated $O_3$ data, in particular the MLS profiles.

For CO, Measurement of Pollution in the Troposphere (MOPITT) thermal infra-red (TIR) V6 total column CO (TCCO) retrievals were assimilated in CAMSRA. These retrievals are most sensitive to CO in the mid and upper troposphere (Deeter et al, 2013) and are retrieved from the TIR band near 4.7 µm. The main improvements compared to the older MOPITT versions used in CIRA (MOPITT V5) and MACC (MOPITT V4) are a correction of a bias in geolocation, improved meteorological data based on the MERRA reanalysis and updated CO a-priori profiles (Deeter et al., 2014). In contrast to MACCRA no IASI
CO retrievals (George et al., 2009; Clerbaux et al., 2009) were assimilated in CAMSRA because using them led to a discontinuity in MACCRA (Inness et al., 2013; Flemming et al., 2017b).

For $NO_2$, tropospheric column retrievals from SCIAMACHY, OMI and GOME-2 were assimilated in CAMSRA. This is an improvement over CIRA (where no $NO_2$ data were assimilated) and MACCRA (where only SCIAMACHY $NO_2$ data were
assimilated). Where possible, new reprocessed data sets were used in CAMSRA. However, due to time constraints it was not possible to acquire and process new observations for all the instruments and for $NO_2$ SCIAMACHY and OMI the data that were already available had to be used. The SCIAMACHY $NO_2$ retrievals used in CAMSRA were the same data version assimilated in MACCRA (KNMI, V1p from 2003-2011, V2 2011-April 2012). The OMI $NO_2$ data were also produced by KNMI (Boersma et al. 2007 and 2011) and consisted of offline DOMINO (V1.0.2) data from October 2004 until 2010, the
offline DOMINO (V2) retrieval for 2011-2012 and NRT DOMINO retrievals (V2) from 2013 onwards. The GOME-2 data were the offline GDP4.8 data produced by the ACSAF/DLR (Valks et al., 2011) until the end of 2016. GOME-2 $NO_2$ retrievals from Metop-A were assimilated from April 2007 onwards and retrievals from Metop-B from January 2013.  In previous studies (Inness et al., 2015) the impact of the assimilation was shown to be small for short lived species like $NO_2$ because at least some of the changes applied to the initial conditions by the analysis were frequently insignificant compared to the prevalent
emissions of nitrogen oxides (NOx) and were lost again in the subsequent forecasts. By assimilating $NO_2$ retrievals from satellites with different overpass times (9:30 local time for GOME-2, 10:00 for SCIAMACHY, 13:30 for OMI) the impact of the assimilation is expected to be increased and the diurnal cycle of $NO_2$ to be better represented.

For aerosols, Collection 6 retrievals of total AOD at 550 nm from the Moderate Resolution Imaging Spectroradiometers
(MODIS, Levy et al., 2018) on board the Aqua and Terra satellites that were produced with the Enhanced Deep Blue (DB) and Dark Target (DT) algorithms over land and a DT over water algorithm over ocean, were used in CAMSRA. The main scientific improvement in the algorithm of Collection 6, compared to the Collection 5 observations used in MACCRA and CIRA, is the introduction of a wind speed dependence over ocean. This addressed the known bias in Collection 5 over mid-latitude oceans, particularly in the Southern Hemisphere.  Various minor changes to the processing were also made in
Collection 6 for maintenance, giving modest improvements (Levy et al. 2013). In addition to MODIS, CAMSRA used retrievals from the Advanced Along-Track Scanning Radiometer (AATSR, Popp et al. 2016) onboard Envisat from 2003 till March 2012.

Averaging kernels were used in the observation operator for the calculation of the model's first-guess fields for CO and $NO_2$
retrievals as described in Inness et al. (2013).





### 3.2 Bias correction

A variational bias correction (VarBC) scheme (Dee and Uppala, 2009) where biases are estimated during the analysis by including bias parameters in the control vector was used for several of the AC data sets. In this scheme, the bias corrections are continuously adjusted to optimize the consistency with all information used in the analysis. VarBC was applied to the

TCO3 retrievals from OMI, SCIAMACHY and GOME-2, with a global constant and solar elevation as predictors, while the partial column SBUV/2, and profile MLS and MIPAS data were used to anchor the bias correction, i.e. were assimilated without correction. Experience from MACCRA had shown that it was important to have an anchor for the $O_3$ bias correction, to avoid drifts in the fields (Inness et al., 2013). The SBUV/2 data were chosen as anchor because they are a high quality reprocessed dataset that covers the whole period of CAMSRA. The MLS and MIPAS profile data were not bias corrected

because experience in MACCRA had shown that the SBUV/2 data could not anchor all the layers of the higher resolved profile data and that drifts in individual layers could lead to problems in the vertical $O_3$ distribution (Inness et al. 2013). Variational bias correction was also applied to OMI $NO_2$ retrievals, again with a global constant and solar elevation as predictors, while SCIAMACHY and GOME-2 $NO_2$ retrievals were used to anchor the bias correction for $NO_2$. This choice was made because SCIAMACHY and GOME-2 generally agree better with the CAMS $NO_2$ fields, while OMI has a larger bias (see Figures S5

and S6 in the supplement) and also suffers from a row anomaly (see supplement) that reduces the number of good data with time. For CO, no bias correction was applied in CAMSRA because data from only one instrument were assimilated and it was not possible to anchor the VarBC. For AOD experience had shown that it was not necessary to anchor the bias correction for the aerosol data and VarBC was applied to both MODIS retrievals and to AATSR. The predictors for AOD were a global constant and the 2-metre wind speed over sea.

The bias correction helps to ensure good time consistency when blending various datasets and adapts to changing biases of the data. An example is shown in Fig. 5 which shows timeseries of monthly mean analysis departures (i.e. observations minus analysis fields) and first-guess departures (i.e. observations minus model first guess) for the four TCO3 retrievals (SCIAMACHY, OMI, GOME-2A & GOME-2B) as well as the applied bias correction. For all four TCO3 data sets the analysis

is drawing to the observations and the standard deviations of the analysis departures are reduced compared to those of the first-guess departures. The plots show that the bias correction (black lines) is different for all instruments, successfully adapts to changes in the data and removes the biases between total column data and the model. OMI data (Fig. 5b) between 2009-2011, for example, show a different behaviour than during the rest of the timeseries with larger departures (due to larger observation values, not shown) and the need for larger bias correction. However, the bias correction successfully accounts for this and the

bias corrected departures are small and stable. The reason for this change is a deterioration in the OMI row anomalies (Torres et al. 2018, see their Figure 1; Schenkeveld et al. 2017). More information about this can be found in the supplement. Thanks to the bias correction such biases are removed, and the bias corrected departures (dotted lines) are small and stable for all 4 instruments.

Monitoring timeseries for all the atmospheric composition datasets assimilated in CAMSRA are shown in the supplement. One important feature to note from the supplement is that SCIAMACHY $NO_2$ (Fig. S5 in the supplement) has much larger positive departures during 2003 than during the rest of the period. This affects the quality of the $NO_2$ analysis during 2003 (see section 4.3 below).

### 4 Results

In this section, analysis fields for $O_3$, CO, $NO_2$ and AOD from CAMSRA are compared with fields from CIRA and MACCRA to highlight some of the improvements in CAMSRA and to point out some of the problems potential users should be aware of.





We concentrate on these four species because they were the ones assimilated in CAMSRA and validation data are available. There are, of course, a lot more species available from CAMSRA that are not covered in this paper. A more thorough validation of the CAMS reanalysis is beyond the scope of this paper and given in validation reports available from the CAMS website (Eskes et al., 2018; available at https://atmosphere.copernicus.eu/sites/default/files/2018-

09/CAMS84_2015SC2_D84.7.1.4_Y14_v1.pdf).

### 4.1 Ozone

We start by looking at TCO3 which is dominated by stratospheric $O_3$ and then look at tropospheric and surface $O_3$ which are more relevant for air quality users. Figure 6 shows the seasonally averaged TCO3 from CAMSRA and the differences between this data set and CIRA and MACCRA. The TCO3 differences between CAMSRA and CIRA are very small (below 2DU,

<1%) with slightly larger differences (5 DU, <3%) in June, July, August (JJA) over Antarctica. CAMSRA TCO3 is slightly higher than CIRA over the Tropical Atlantic in all seasons and in NH midlatitudes in JJA, lower over NH midlatitudes during December, January, February (DJF) and March, April, May (MAM), and lower over SH midlatitudes in MAM and JJA. The differences between CAMSRA and MACCRA are larger, with CAMSRA lower than MACCRA everywhere (up to -10 DU, <5%).

To assess if the differences seen between CAMSRA and the older reanalyse are an improvement we compare TCO3 from the reanalyses with independent, i.e. not used in the analysis, Dobson sun photometer measurements (Fig. 7) obtained from the World Ozone and Ultraviolet Radiation Data Centre (WOUDC). The Dobson data are well calibrated with a precision of 1% (Basher, 1982). This comparison shows that MACCRA has the largest (positive) biases relative to these data and that

CAMSRA agrees better with the independent observations in all areas. CAMSRA has smaller biases than the other two reanalyses in all areas, except in the Tropics after 2013 when CIRA has a smaller bias. CAMSRA and CIRA are very close from 2003-2012, but diverge more from 2013 onwards, when the version of the MLS profiles used in CIRA changed from V2 to NRT V3.4 (Flemming et al., 2017b). In these later years, CAMSRA is generally better than CIRA except in the Tropics. The largest biases for CAMSRA (up to 25 DU) are found over Antarctica during the ozone hole season after 2013. The figure

shows that there is no noticeable impact during 2009-2011 when degraded OMI observations were assimilated (Fig. 5), illustrating the success of the variational bias correction (Section 3) got the TCO3 data. We see that for TCO3 CAMSRA is clearly a better product than the older reanalyses.

While it is relatively easy to reproduce a good TCO3 field by assimilating TCO3 data, reproducing the vertical structure of the

$O_3$ field is more difficult and the CAMS system had problems with this in the past (Flemming et al., 2011). We therefore also carry out a comparison against independent ozone sondes to evaluate the vertical structure of model biases in the troposphere and stratosphere. The ozone sonde data used for the validation were acquired from a variety of data centres: WOUDC, Southern Hemisphere ADditional OZonesondes (SHADOZ), Network for the Detection of Atmospheric Composition Change (NDACC), and campaigns for the Determination of Stratospheric Polar Ozone Losses (MATCH). The precision of

electrochemical concentration cell (ECC) ozone sondes is on the order of ±5% in the range between 200 and 10 hPa, between −14% and +6% above 10 hPa, and between −7% and +17% below 200 hPa (Komhyr et al., 1995). Larger errors are found in the presence of steep gradients and where the ozone amount is low. The same order of precision was found by Steinbrecht et al. (1998) for Brewer–Mast sondes. Mean relative difference between the three reanalyses and ozone sondes and the standard deviations of the biases are shown in Fig. 8 for the Globe, Arctic, NH midlatitudes, Tropics, SH midlatitudes and Antarctic.

For MACCRA the average is only for the period 2003-2012. In general, CAMSRA agrees to with 10% with the sondes. The best agreement between the reanalyses and the sondes is found in the stratosphere where the assimilated $O_3$ data constrain the analyses well. Differences between the reanalyses are larger in the troposphere where the impact of the assimilation is smaller



(Inness et al., 2015) and differences in the chemistry schemes, emissions and transport become more important. CAMSRA and CIRA agree well above about 200-100 hPa, while MACCRA overestimates $O_3$ in all areas above about 15 hPa. While this overestimation of upper stratospheric and mesospheric $O_3$ in MACCRA will not affect the TCO3 bias, ozone in this region is important for radiative transfer and the associated heating rates. A smaller bias in this region will make CAMSRA a better

dataset to be used as climatology in e.g. radiation schemes or radiance observation operators. CAMSRA has larger $O_3$ values than CIRA in the troposphere which leads to an increased bias with respect to the sondes in the Tropics, but smaller biases in the other areas. Near the surface CAMSRA has a positive bias. The largest differences between the reanalyses in the troposphere are found in the Tropics. Here MACCRA underestimates $O_3$ in the mid and upper troposphere with mean biases up to -30%, but absolute differences are small because $O_3$ values in the tropical upper troposphere and lower stratosphere are

low. MACCRA also has a large negative bias near the surface in the Arctic and Antarctic. Here, improvements to the background error statistics (Inness et al., 2015), in particular to the vertical correlations of the background errors, led to big improvements in CIRA and CAMSRA compared to MACCRA.

The profile plots have shown that the largest relative differences between the three reanalyses are found in the troposphere.

Therefore, Fig. 9 looks at timeseries of the modified normalized mean bias (MNMB) of reanalysis $O_3$ minus ozone sondes in the free troposphere (layer between 750-300 hPa) to assess these differences in more detail. Figure 9 confirms that MACCRA has the largest bias with respect to the sondes and shows a different behaviour between mid-2004 and the end of 2007 than during the other years, particularly noticeable in the Arctic, NH midlatitudes and Antarctic. This was documented in Inness et al. (2013) and was the result of using VarBC for MLS data in MACCRA during the period August 2004 till December 2007.

CAMSRA is a much improved and temporally more consistent dataset than MACCRA. CAMSRA also has a smaller bias than CIRA in all areas, apart from the NH midlatitudes during 2005-2009. CAMSRA has larger $O_3$ values than CIRA in the free troposphere, so that CAMSRA shows a small positive bias and CIRA a small negative bias, which was also seen in the $O_3$ profile plots (Figure 16). CIRA and MACCRA have larger biases than CAMSRA in 2003 which could be the result of assimilating GOME $O_3$ profiles during the first 5 months of 2003 in CIRA and MACCRA, but not in CAMSRA (because it

was found to lead to a degradation in the CAMS $O_3$ analysis, not shown). It was shown previously (Inness et al., 2013; Flemming et al., 2011; Lefever et al., 2015) that it is important in the CAMS system to assimilate height resolved $O_3$ data, like MLS profiles, to obtain a good vertical structure of the $O_3$ analysis and this is confirmed by Fig. 9 as all areas apart from the NH midlatitudes show larger biases from the end of March to the beginning of August 2004 when no $O_3$ profile data were assimilated (Figure 4). The biases in the Arctic and Antarctic regions are larger during 2003 than during the other years. This

seems to be related to the degraded quality of the NRT SCIAMACHY and MIPAS data used during 2003 (Figures 5 and S1 in the supplement). The user has to be aware of these problem periods.

There is a change in the bias behaviour from January 2013 onwards in CAMSRA and CIRA, particularly in the Antarctic and Arctic, where biases increase compared to the earlier years and show a seasonally varying behaviour. This must be the result of changes in the observing system, as the model does not change and is currently under investigation. Tests so far suggest

that it is not due to the change from 13L to 21L SBUV/2 data or due to the loss of Envisat in April 2012. It might be related to the use of GOME-2 data, but further tests are necessary to establish this for sure. The same seasonally varying biases are also found in the CAMS real-time system (not shown) from 2013 onwards.

To finish the $O_3$ validation we look at surface ozone data. Figure 10 shows timeseries of MNMB of the reanalyses with respect to ground-based surface observations from the WMO's Global Atmosphere Watch (GAW) programme (e.g., Oltmans and

Levy, 1994; Novelli and Masarie, 2014) averaged globally and for Europe. The GAW observations represent the global background away from the main polluted areas. Detailed information on GAW can be found in GAW reports No. 209 (2013) (http://www.wmo.int/pages/prog/arep/gaw/gaw_home_en.html). GAW $O_3$ data have a precision of ±1 ppbv (Novelli and



Masarie; 2014). In the global mean CAMSRA agrees with the surface data to within 10% for most years. The biases are generally negative during the first half of the year and positive during the second. MACCRA has larger negative biases after 2008, and CIRA larger negative biases from 2003-2012. Surface $O_3$ in CAMSRA is higher than in CIRA so that the global mean biases during boreal spring are smaller, but the positive global mean biases during boreal summer larger. After spring

2013 CAMSRA and CIRA are very close. During 2003 CIRA and MACCRA have a considerably larger bias than CAMSRA. This is also seen over Europe and North America and was also seen in ozone in the free troposphere (Figure 9). In Europe CAMSRA has biases between -40 and +10%. All three reanalyses show an underestimation of surface $O_3$ during boreal spring and better agreement with the observations during summer when the bias is positive. The negative spring time bias is a known problem in the CAMS system and is generally smaller in CAMSRA than CIRA. The largest negative bias in CAMSRA is seen

during 2004 (when no $O_3$ profile data were assimilated). Overall, CAMSRA is the most consistent data set with time.

In summary, it can be said that for $O_3$ CAMSRA is temporally more consistent than the older reanalyses and has smaller biases compared to independent observations. The comparisons also show that it is not advisable to concatenate the older reanalyses with more recent years from CAMSRA, because the datasets are too different, and that users should use only data from

CAMSRA if they are interested in the complete period from 2003-2016. There are some periods with slightly degraded quality (bigger biases) that the user should be aware of. These include the Arctic and Antarctic free troposphere during 2003 because MIPAS and SCIAMACHY data or poorer quality were assimilated, the period between the end of March and the beginning of August 2004 when no profile data were available for assimilation, and a change in bias after 2013 that is still under investigation. The underestimation of surface O3 seen in the CAMS system in the NH during boreal spring is reduced in

CAMSRA compared to the older reanalyses.

**4.2 Carbon Monoxide**

Next, we look at CO fields from the reanalyses and compare them with independent observations. Figure 11 shows the seasonally averaged TCCO fields from CAMSRA and the differences between this data set and CIRA and MACCRA. The

TCCO differences between CAMSRA and CIRA are small (below 0.1 x$10^{18}$ molec/cm$^2$, <5%) with CAMSRA generally lower than CIRA, apart from African biomass burning areas south of the equator in JJA and parts of SE Asia in DJF and MAM. The largest relative differences (of up to 15%, not shown) are found over the tropical oceans where background values are small. The differences between CAMSRA and MACCRA are larger. CAMSRA is lower than MACCRA over the oceans (0.1-0.2 x$10^{18}$ molec/cm$^2$, relative differences mainly < 15%) and much higher over biomass burning areas, e.g. South America, Africa,

South-East Asia, Indonesia, Australia in DJF and boreal fires in Siberia in MAM and JJA with differences up to 0.5 x$10^{18}$ molec/cm$^2$ (corresponding to maximum relative differences of up to 30% over Indonesia). These difference plots show that the choice of fire emissions used in the reanalysis has a large impact on the TCCO field. In MACCRA these came from GFED (van der Werf et al., 2010) for the period 2003-2008 and GFAS v1.0 from 2008-2012 (Kaiser et al., 2012), while in CAMSRA and CIRA GFAS V1.2 was used throughout from 2003-2016 (see Table 1 and Fig. 3). As for $O_3$, the differences between the

reanalyses are too large to allow the user to concatenate recent years from CAMSRA with earlier years from the other reanalyses.

To validate CO from the reanalysis with independent observations, in Fig. 12 we first compare our data with observations from Total Carbon Column Observing Network stations (TCCON, GGG2014 data, Wunch et al., 2011, see www.tccon.caltech.edu)

at six sites covering latitudes from the Arctic to Australia (see Table 3). The TCCON stations measure the column-averaged dry molar fraction CO amount (XCO) and have an absolute accuracy of about 4% (Wunch et al., 2010). Figure 12 shows very good agreement of CAMSRA with the independent observations, in particular, for the year-to-year variability. The mean bias





and standard deviations of the three reanalyses against the TCCON data are given in Table 3 and show that the mean biases and standard deviation in CAMSRA are reduced at all stations compared to MACCRA. CAMSRA is slightly lower than CIRA with smaller biases and standard deviation at all stations except Bremen and Sodankyla. Particularly in the tropics and SH the biases and standard deviations are much reduced in CAMSRA. CAMSRA captures the seasonal cycle well at all stations.

Especially the summer minimum (e.g. boreal summer in NH, austral summer in SH) is better captured in CAMSRA than in CIRA. CAMSRA underestimates XCO at the NH stations Ny-Ålesund, Sodankyla, Bremen and Park Falls (biases < -2 ppb) and overestimates it in the Tropics (Izaña, < 5 ppb; Darwin, < 2 ppb) and in the SH (Lauder, <1 ppb). CIRA slightly overestimates XCO in the NH (< 3ppb) and has a larger positive bias than CAMSRA in the Tropics and SH (up to 8 ppb).

To also assess the vertical structure of the CO analysis fields, in Fig. 13 we compare model fields with CO profiles from MOZAIC (Measurements of OZone, water vapour, carbon monoxide and nitrogen oxides by in-service AIrbus aircraft) and IAGOS (In-service Aircraft for a Global Observing System) observations from instruments mounted on commercial aircraft. The MOZAIC/ IAGOS CO data have an accuracy of ± 5 ppbv, a precision of ± 5 % and a detection limit of 10 ppbv (Nédélec et al, 2003).  We use CO profiles obtained during take-off and landing to evaluate the CO reanalysis fields. The profiles at the

NH mid latitude airports (Frankfurt, Eastern US, Japan) show that all three reanalyses underestimate CO in the free troposphere, but agree to within 10% with the aircraft data. A larger underestimation is found in the boundary layer. Here, MACCRA has the largest negative bias. This underestimation in MACCRA was noted previously (Inness et al., 2013) and led to a modification of increased winter time road traffic emissions over North America and Europe in the MACCity emissions (Stein et al., 2014). These modified emissions are used in CAMSRA and CIRA. CAMSRA and CIRA are generally closer to

each other in the lower troposphere than to MACCRA. This area is less impacted by the assimilated MOPITT TIR retrievals that have the largest sensitivity to CO in the mid troposphere (Deeter et al., 2013) and more by emissions and differences in the chemistry schemes, which are more similar in CAMSRA and CIRA than in MACCRA. In the upper troposphere CAMSRA has the lowest mean bias while CIRA and MACCRA overestimate CO above about 300 hPa. At Windhoek all reanalyses underestimate the aircraft data. Here CAMSRA and MACCRA have a smaller bias than CIRA below 650 hPa, but CIRA has

a smaller bias above 500 hPa.  Over SE Asia all reanalyses show a large underestimation in the boundary layer with MACCRA having the largest bias of up to -35%. In the free troposphere all reanalyses underestimate CO, but have a smaller bias than near the surface. MACCRA has the smallest bias in the free troposphere (biases of less than -5% between 650-400 hPa). This could be the result of assimilating IASI TCCO data (George et al.; 2009; Clerbaux et al.; 2009) in MACCRA in addition to MOPITT. Like MOPITT, IASI CO retrievals are most sensitive to CO in the mid troposphere and could add an extra constraint

on CO here when more observations are being assimilated, as IASI has a better coverage than MOPITT (e.g. Barré et al., 2015). Over Indonesia CAMSRA and CIRA have smaller biases than MACCRA below 700 hPa. This is likely due to differences in the fire emissions. At Windhoek, SE Asia and Indonesia CAMSRA and CIRA overestimate surface CO. This overestimation is also seen in comparison with GAW surface CO data at Cape Point (not shown).

Next, we look at surface CO data. Figure 14 shows maps of mean biases of surface CO against GAW observation. The data are averaged over the period 2003-2016 for CAMSRA and CIRA and 2003-2012 for MACCRA. The uncertainty of GAW CO data is between 2 ppbv for marine boundary layer sites and 5 ppbv for continental sites that are influenced by regional pollution (WMO, 2010).  The biases in CAMSRA and CIRA are less than 10% for many stations, with slightly larger positive biases for some North American and slightly larger negative biases for some European stations. MACCRA has larger negative biases

over North America and Europe. CAMSRA shows a larger positive bias than the other two reanalyses at the Indonesian station of Bukit Koto Tabang. Looking at a timeseries at this location (Figure 23d) we see that the station is strongly influenced by high CO events during years with intense biomass burning (2004, 2006, 2014, 2015) with the largest peaks in 2014 and 2015



when CAMSRA is higher than CIRA. This is after the end of MACCRA which only covered the period from 2003-2012. It has to be assess if this is the result of too large GFAS emission factors for CO.

Figure 15 shows timeseries of monthly mean CO surface biases of the reanalyses with respect to GAW observations averaged

over Europe and timeseries of absolute surface CO values at the Arctic Alert station and the Indonesian Bukit Koto Tabang station. The agreement of MACCRA with GAW CO data over Europe (Fig. 23a) is worse than for the other two reanalyses with a large underestimation during boreal winter. This bias was already documented in Inness et al. (2013) and Flemming et al. (2017b). This negative bias of MACCRA increases after April 2008 when the assimilation of IASI CO retrievals started in MACCRA (see Inness et al., 2013) and is particularly pronounced at high northern latitudes (e.g. timeseries at Alert, Fig. 23b).

In summary, CO from CAMSRA has a good seasonal cycle and captures the interannual variability observed by TCCON data well. CAMSRA has smaller biases relative to TCCON data than the older reanalyses at most stations. There is a low bias with respect to IAGOS aircraft data in the lower troposphere over NH midlatitudes, SE Asia and Indonesia. This is a persistent feature of the CAMS system and more work is needed to assess if it is a model problem or due to problems with the emissions.

CAMSRA is more similar to CIRA than MACCRA because of differences in the emissions, chemistry schemes and assimilated data. It is therefore not possible to use a climatology based on the MACCRA data and recent years from CCAMSRA to e.g. calculate anomalies. CAMSRA generally agrees better with GAW surface observations than MACCRA, but has larger biases relative to GAW over Indonesia.

**4.3 Nitrogen Dioxide**

The final reactive gases species discussed in this paper is $NO_2$. Validation of $NO_2$ with independent observations, especially surface observations, is difficult because of the short lifetime and large, orders of magnitude, variability of the concentrations. First, we compare the seasonally averaged total column $NO_2$ fields from CAMSRA and CIRA in Fig. 16. Because the IFS(CB05) scheme is a tropospheric scheme, the total column $NO_2$ fields from CAMSRA and CIRA are basically tropospheric

columns. MACCRA is not shown in Fig. 16 as it used a chemistry scheme that included the stratosphere and has very different total column $NO_2$. Figure 16 shows that CAMSRA has a realistic $NO_2$ distribution with high $NO_2$ columns in the NH in areas affected by anthropogenic emissions and also in boreal and tropical biomass burning areas. The largest $NO_2$ values are found in DJF in the regions of anthropogenic pollution when emissions are largest and the $NO_2$ lifetime longest. In Africa, a realistic seasonal cycle is found with a maximum in the Sahel in DJF, and a maximum south of the equator in JJA related to the

seasonality of biomass burning. $NO_2$ columns over South America are smaller than over Africa, with the largest values found in SON because deforestation fires and agricultural fires mainly occur south of 10°S during August-October with peak in September. Over Indonesia the largest $NO_2$ columns are also seen in SON and are dominated by contributions from the 2015 fires which show up as the largest spike of the GFAS emissions in Asia (see Fig. 3).

Differences between CAMSRA and CIRA are due to model changes, but also due to the assimilation of the tropospheric column $NO_2$ retrievals from SCIAMACHY, OMI and GOME-2 in CAMSRA (see Table 2). No $NO_2$ retrievals were assimilated in CIRA. CAMSRA generally has larger $NO_2$ columns than CIRA in areas affected by biomass burning. During JJA and DJF, CAMSRA is higher than CIRA in most of the Tropics, while the differences are negative at high latitudes. Notable exceptions are the much lower values in CAMSRA over the Arabian Peninsula (> -50%). This reduction is not due to the assimilated data,

but due to model changes, i.e. the coupling with aerosol in the chemistry scheme (see section 2.1) that leads to faster NOx removal and reduces the positive bias seen before in this area when evaluating against satellite $NO_2$ observations (not shown). In midlatitudes of both hemispheres the differences between CAMSRA and CIRA are mainly small and positive in JJA and



SON. In DJF and MAM there are again positive differences in the Tropics, but these are more confined to South America, Africa and Indonesia than in JJA and SON. Positive differences are also found over India and SE Asia, while most extratropical areas show negative differences (apart from the areas affected by boreal fires). There are large differences over the poles in relative terms, but these are small in absolute terms as $NO_2$ columns are small here. The fact that the largest relative differences

in the NH (outside the polar regions) are seen during DJF and MAM suggests that this is at least partly due to the impact of the assimilation of the satellite data. While the impact of the $NO_2$ assimilation is generally small because of the short lifetime of $NO_2$, it was found to have a larger impact during winter and spring when the lifetime is longer than during summer (see supplement Fig. S5 and S6 and Inness et al., 2015). Furthermore, by assimilating $NO_2$ retrievals from satellites with different overpass times (9:30 local time for GOME-2, 10:00 for SCIAMACHY, 13:30 for OMI) the impact of the assimilation is likely

to be increased.

It is difficult to find independent $NO_2$ data for validation which are representative for the grid box size of the CAMSRA global reanalysis. We therefore use the following two datasets for validation. (1) A satellite based tropospheric column $NO_2$ dataset and (2) surface $NO_2$ measurements from selected GAW stations in Europe. GAW stations aim to have an uncertainty of about

3% for monthly mean data (Penkett et al., 2011). As the number of GAW stations measuring $NO_2$ is small and drops considerably with time during the period of interest, it is not meaningful to look at timeseries of area means. We therefore restrict our validation to four European GAW stations that have observations for most of the period from 2003-2016. The dataset (1) is produced by the University of Bremen based on SCIAMACHY/Envisat $NO_2$ satellite retrievals (IUP-UB v0.7, before April 2012) (Richter et al., 2005) and GOME-2/MetOp-A $NO_2$ satellite retrievals (IUP-UB v1.0, from April 2012 until

the end of 2016) (Richter et al., 2011). The retrieval product used for validation is a different one than the SCIAMACHY and GOME-2 $NO_2$ retrievals that are assimilated in CAMSRA (a retrieval product produced by the ACSAF, see Table 2). Despite the retrievals being based on the same level 1 spectral irradiance data, the retrieval procedures are completely independent, from the spectral fit to the assumptions made on the a-priori used for the air mass factor calculations. In the absence of other independent validation data for tropospheric $NO_2$ columns, they can still provide a critical evaluation of the model performance

on a global scale. The satellite data are always taken at the same local time, roughly 10:00 LT for SCIAMACHY and 09:30 LT for GOME-2, and at clear sky only. Model data are vertically integrated, interpolated linearly in time to the observation time of SCIAMACHY (which is expected to lead to minor uncertainties when comparing to GOME-2 observations in Figure 25) and then sampled spatially to match the satellite data. Model data were treated with the same reference sector subtraction approach as the satellite data. Uncertainties in $NO_2$ satellite retrievals are large and depend on the region and season. Winter

values in mid and high latitudes are usually associated with larger error margins. As a rough estimate, systematic uncertainties in regions with significant pollution are on the order of 20% – 30%.

Figure 17 shows timeseries of tropospheric column $NO_2$ from the Bremen satellite data set (0.5º x 0.5º), CAMSRA, CIRA and MACCRA averaged over Europe and East-Asia for the period from 2003 to 2016. The figure illustrates that, while the

seasonality of $NO_2$ (with low values during summer and high values during winter) is captured in both areas, there is generally an underestimation of the seasonal cycle, mainly due to an underestimation of the winter time maximum. This underestimation could be related to an underestimation of anthropogenic emissions or uncertainties in the photochemistry of the models and is particularly pronounced over East Asia. Over Europe the differences between the three reanalyses are small; over East Asia they are larger. Over East Asia in 2003, the CAMS reanalysis shows a strong variation of values from one month to the next

and fails to reproduce the observed seasonality. This is due to assimilating SCIMACHY $NO_2$ data of degraded quality during 2003 in CAMSRA (see supplement Fig. S5a). The Bremen data set shows an increase of the wintertime maximum $NO_2$ values over East Asia until 2014 and a decrease in the later years. This behaviour is reproduced better in CAMSRA than in CIRA and MACCRA, though the maximum values are still underestimated. This improvement is the result from assimilating more $NO_2$



satellite data, in particular data from satellites with different overpass times, in CAMSRA. It is not seen in a control run without data assimilation (not shown). However, the magnitude of the positive trend up to 2014 and of the negative trend in the recent years is still underestimated by CAMSRA and the observed decrease between 2013 and 2014 is not reproduced by the three reanalyses.

Figure 18 shows timeseries of surface $NO_2$ from CAMS, CIRA and MACCRA with GAW surface observations at four European stations. Overall, the reanalyses reproduce the observed mean values and the seasonal variability well. At Leba (at the Baltic coast) all three reanalyses capture the annual cycle well with high $NO_2$ concentrations during the winter and lower concentrations during the summer, and the three reanalyses are quite similar. MACCRA underestimates the summertime

minimum more than the other two reanalyses. At Jarczew (Poland) both CAMSRA and MACCRA capture the low summertime $NO_2$ values better than CIRA which has a positive bias during summer, while the wintertime $NO_2$ maxima are more similar in the three reanalyses. Overall, CAMSRA agrees best with the GAW data here. At Hohenpeissenberg in Southern Germany all reanalyses underestimate the summer minimum and struggle to capture some of the high winter values. CIRA underestimates the winter maximum values most while CAMSRA and MACCRA agree better with the observations during winter, especially

during the first half of the timeseries. At Payerne (Switzerland), MACCRA strongly underestimates the GAW observations, while CAMSRA and CIRA capture the annual cycle reasonably well, in particular the summer minimum.

In summary, we find that CAMSRA shows some improvements in the tropospheric $NO_2$ column (compared to a dataset based on SCIMACHY and GOME-2A data) relative to the older two reanalyses, especially over East Asia where the assimilation of

(more) $NO_2$ retrievals reduces the bias between the reanalysis and the data. However, the tropospheric $NO_2$ columns are still underestimated in CAMSRA over East Asia and Europe, particularly the wintertime maxima. This is a longstanding problem of the CAMS system and it is hoped that work which has started to include an emission inversion capability in the CAMS system will improve this in the future. We find changes in $NO_2$ compared to CIRA (particularly over the Arabian Peninsula), which are the result of coupling with aerosol in the CAMSRA chemistry scheme that leads to faster NOx removes and a

reduced positive bias in those areas. Compared to European GAW surface observations CAMSRA reproduces the monthly mean values and the seasonal variability well with some underestimation of wintertime maximum values.

**4.4 Aerosols**

The final validation section looks at aerosol fields from the reanalyses. Several model changes were included in the version of

the IFS used to produce CAMSRA to address issues identified in CIRA and this has a large impact on the aerosol speciation. Figure 19 shows the mean AOD over the period from 2003 to 2016 from CAMSRA and the differences between this dataset and CIRA and MACCRA (only for 2003-2012). Also shown are the mean and differences for the individual aerosol components (sea salt, desert dust, sulphate, organic matter and black carbon). There is a considerable change in the aerosol composition in CAMSRA. Relative to CIRA, CAMSRA shows a reduction in desert dust, sulphates and black carbon in the

SH, compensated by an increase in sea salt, organic matter and black carbon in the NH. Compared to MACCRA there is a reduction in sea salt, desert dust, sulphate and black carbon in the SH, and an increase in organic matter and black carbon in the NH. Too much sulphate was a known problem of CIRA, where it was the dominant species contributing to AOD in regions away from the main aerosol emissions (Flemming et al., 2017b). This resulted partly from the mis-speciation of analysis increments mentioned in Section 2.3. This is significantly improved in CAMSRA by model changes, accompanied by a large

increase in organic matter in polluted regions from the introduction of a representation of anthropogenic SOA as described in Section 2.1.1, which was missing from the earlier reanalyses.




Total AOD in CAMSRA is reduced over most land areas and the Arctic Ocean (Fig. 19); however, there are increases over most of the tropical ocean and non-desert tropical land regions, in particular SE-Asia, India, Indonesia and parts of tropical South America and Africa. The largest absolute reduction is found in desert areas (North Africa, Middle East, Gobi) where CAMSRA is up to 0.2 lower than CIRA, where model changes led to a reduction in desert dust. The reduction of AOD seen

in the NH comes from the reduction in sulphate. Differences of the total AOD between CAMSRA and MACCRA are larger than between CAMSRA and CIRA with CAMSRA considerably lower than MACCRA everywhere except the Sahara, tropical South America and parts of SE Asia.

The AOD at 550 nm from the reanalyses is evaluated with observations of the AErosol RObotic NETwork (Aeronet, Figures

28-30) Version 3 Level 2 data. Aeronet is a network of about 400 stations measuring spectral AOD with ground-based sun photometers (Holben et al., 1998). The stations are mostly located over land, with a high number of stations situated in North America and Europe. The global number of stations contributing observations for the evaluation increased from about 60 in 2003 to about 300 in 2016. Figure 20 shows maps of the mean biases from the three reanalyses against Aeronet. CAMSRA has the smallest mean bias at most locations, while CIRA shows larger positive biases over North America, Australia and

desert areas (North Africa, Middle East, Gobi) and a larger underestimation in India and SE Asia. MACCRA has even larger positive biases in North America and larger positive biases in Europe and the Mediterranean. Figure 20 shows that in CAMSRA there are some hot spots around outgassing volcanoes (in particular Mauna Loa and Mexico City) with high analysis AOD values that degrade the global average bias. These hotspots are dominated by sulphate, and are a side effect of possibly erroneous model treatment of diffuse volcanic emissions, i.e. the model-resolution orography not resolving the height of the

volcanoes and therefore not being representative of the measurement sites with respect to the volcanic plumes. It is also possible that there are errors on how quickly aerosol is formed from diffuse outgassing sources. The volcanic emissions have been unmasked by recent enhancements to the aerosol $SO_2$ oxidation scheme which improve aerosol on the global scale. When calculating global mean statistics it is advisable to exclude those two stations as unrepresentative.

Figure 21 shows timeseries of monthly mean biases from the three reanalyses against Aeronet for several areas and Figure 22 shows global correlation coefficients. As explained above, Mauna Loa and Mexico City were excluded from these statistics. As already seen in Figure 20, CAMSRA generally has the smallest bias with respect to the Aeronet data and the largest correlation coefficient (Figure 22). It shows a good consistency throughout the time period from 2003-2016 while MACCRA shows an increasing positive bias with time in Europe and North America. CIRA also shows increasing positive biases with

time in North America, particularly from 2013 onwards, and a change in biases in Europe, from negative at the beginning of the timeseries, to positive at the end. It still has to be assessed if this improvement is due to model changes or a better representation of the emissions in CAMSRA. There is a change to slightly higher AOD in CAMSRA (biases more positive in the global mean and in particular over Europe and North America) that seems to coincide with the loss of AATSR data in April 2012. Over SE-Asia all reanalyses have a negative bias with CAMSRA having the smallest and CIRA the largest. In the

NH, the bias changes with season and is largest (positive) during the summer months.

We do not have observations to validate the individual aerosol components, but the simulated aerosol size distribution and implicitly the aerosol composition may be validated to first order by using the wavelength dependent variation in AOD. It is expressed as Ångström exponent (AE), with higher Ångström exponents indicative of smaller particles and dominance of

sulphate and organic aerosols. AE is little dependent on wavelength itself. We compute it here from AOD@440 and AOD@870 nm, except in CIRA, where only AOD@550 and AOD@670 nm were available. Figure 23 shows the temporal evolution of simulated and observed mean AE, as well as the correlation. CAMSRA and CIRA show less variability compared to the observations, overestimating mostly the Ångström exponent (5-20%). Overestimation appears mainly in late spring, indicating



possibly too little coarse dust. The bias is, however, considerably smaller than for MACCRA, the latter having a significant low bias. Total AOD is composed of less dust in CAMSRA and CIRA compared to MACCRA. This may explain the higher overall Ångström values in the CAMS reanalysis. Temporal-spatial correlation in Fig. 23 is higher in winter in the CAMSRA and indicates partially better tracing of aerosol size and implicitly composition variability than in both CIRA and MACCRA

reanalysis.

In summary, there has been a large change in aerosol composition in CAMSRA compared to the previous reanalysis, making it impossible to compare aerosol species from CAMSRA with climatologies built from CIRA or MACCRA. There is a pronounced reduction in sulphate in CAMSRA which was too high in CIRA. More work is needed to validate the individual

aerosol components against independent observations. CAMSRA total AOD shows a reduced bias against Aeronet observations and a better temporal consistency, while the older reanalyses show biases that increase with time over North America and Europe. CAMSRA shows too high AOD values around outgasing volcanoes (Mauna Loa, Mexico) and we recommend to exclude those locations as unrepresentative when calculating global mean statistics.

### 5 Conclusions

The Copernicus Atmosphere Monitoring Service (CAMS) has produced a new reanalysis dataset of atmospheric composition, referred to as CAMSRA in this paper. This reanalysis currently covers the years 2003-2016 and will be extended in the future by adding one year each year. It was produced by assimilating satellite retrievals of $O_3$, CO, $NO_2$ and AOD from various sensors in ECMWF's Integrated Forecast System (IFS). The new CAMS reanalysis builds on the experience gained during the production of the earlier MACC reanalysis (MACCRA) and CAMS interim reanalysis (CIRA). Great care had been taken to

ensure that the emission datasets used in CAMSRA were consistent in time and that consistent anthropogenic, biogenic and biomass burning emissions were used in the aerosol and chemistry modules. Furthermore, a newer (and therefore better) version of the IFS was used and new, reprocessed data sets for assimilation were acquired as far as possible. Variational bias correction was applied to some of the $O_3$, CO, $NO_2$ and AOD data to ensure good temporal consistency when blending the various datasets. Known problems from earlier reanalyses were avoided, e.g. issues with the bias correction of MLS data in

MACCRA that led to drifts in the ozone field, and a better time consistency in the CO field of CAMSRA than of MACCRA was obtained by assimilating data from only one instrument, i.e. MOPITT. CAMSRA therefore shows a more temporally consistent performance than the previous two reanalyses and has mostly smaller biases with respect to independent observations in most areas for $O_3$, CO, $NO_2$ and AOD.

The validation results presented in this paper have shown that mean TCO3 fields from CAMSRA and CIRA are similar and agree to within 1% when averaged over the period 2003-2016. The differences between CAMSRA and MACCRA are larger, but are still within 5%. All reanalyses have small positive biases with respect to Dobson TCO3 observations, with MACCRA having the largest biases and CAMSRA the smallest. Agreement with ozone sondes is within 10% in the long term global mean. The reanalyses agree well in the stratosphere and have larger differences in the troposphere. CAMSRA agrees better

with ozone sondes above 15 hPa than MACCRA which overestimates $O_3$ there. This makes CAMSRA a better dataset to be used as climatology in e.g. radiation schemes or radiance observation operators. CAMSRA and CIRA agree better with ozone sondes in the tropical mid to upper troposphere than MACCRA which shows a large underestimation here (-30%). $O_3$ from CAMSRA is more consistent in time than MACCRA because the variational bias correction applied to MLS $O_3$ retrievals during some of MACCRA led to drifts in the $O_3$ field, particularly noticeable in the troposphere in MACCRA (Inness et al.,

2013). CAMSRA shows a smaller underestimation of surface $O_3$ in the NH during boreal spring than the previous reanalyses. We note, that there is an increased seasonally varying tropospheric ozone bias in CAMSRA after 2013, particularly in the




Antarctic and Arctic. The reason for this bias is still being investigated and the same bias is also found in the CAMS real-time ozone analysis. There are larger ozone biases in all three reanalyses between March and August 2004 when no $O_3$ profile data were available for assimilation than during the rest of the period, and larger biases during 2003 when MIPAS and SCIAMACHY $O_3$ data of poorer quality were assimilated.

For CO, CAMSRA shows good agreement with TCCON observations with small biases and a good representation of the seasonal cycle and inter-annual variability. CAMSRA has the smallest bias out of the three reanalyses with respect to most of the TCCON stations looked at in this paper, with a small negative bias in the NH and a small positive bias in the Tropics and SH. Especially in the Tropics and SH the biases in CAMSRA are much reduced compared to CIRA and MACCRA.

Comparisons with IAGOS aircraft data show an underestimation of CO in the free troposphere in the NH (<10%) with larger underestimation in the lower troposphere. This underestimation is similar in CAMSRA and CIRA, while MACCRA has larger negative biases in the NH lower troposphere. CAMSRA also has smaller biases with respect to GAW surface CO than MACCRA. The choice of fire emissions has a large impact on the TCCO field, and the largest differences between CAMSRA and MACCRA are seen in biomass burning areas, because different fire emission datasets were used in these reanalyses. CO

from CAMSRA is more consistent in time than MACCRA, which showed some changes in the CO field because the assimilation of IASI CO was started in MACCRA in 2008 (Inness et al., 2013), while in CAMSRA and CIRA only TCCO from MOPITT was assimilated.

For $NO_2$, data from more instruments were assimilated in CAMSRA (SCIAMACHMY, OMI & GOME-2AB) than in CIRA

(none) and MACCRA (SCIAMACHY only). This led to differences between the datasets but the limited amount of independent validation observations for $NO_2$ made it difficult to assess these differences. The seasonal cycle of the tropospheric $NO_2$ columns is underestimated over East Asia and to a smaller extent over Europe by all three reanalyses compared to a tropospheric column $NO_2$ dataset based on SCIAMACHY and GOME-2A data. CAMSRA shows the smallest bias out of the three reanalyses over East Asia, suggesting that the assimilation of several $NO_2$ satellite retrievals improves the $NO_2$ analysis.

However, the comparison also showed that the quality of the $NO_2$ analysis in CAMSRA was degraded during 2003 because of the reduced quality of the assimilated SCHIAMACHY $NO_2$ data (Fig. S5a in the supplement) during that time. Compared to European GAW $NO_2$ surface observations, the reanalyses reproduced the observed mean values and the seasonal variability well, but again showed an underestimation of high winter time values. More work is needed to validate the $NO_2$ fields from CAMSRA thoroughly and to assess if the wintertime underestimation is due to shortcomings of the model or the emissions.

Total AOD values in CAMSRA are reduced compared to CIRA and MACCRA in many areas, but increased over India and SE Asia and agree better with Aeronet total AOD. AOD in CAMSRA is more consistent in time than in CIRA and MACCRA, especially over Europe and North America where CIRA and MACCRA show an increasingly positive bias with time. There are large differences in aerosol speciation (which is less constrained by the assimilated AOD observations) between CAMSRA,

CIRA and MACCRA. Relative to both the earlier reanalyses, CAMSRA shows a reduction in desert dust, sulphates and black carbon in the SH, compensated by an increase in organic matter and black carbon in the NH. The reduction in sulphate globally is particularly strong relative to CIRA, where its contribution was overestimated (Flemming et al., 2017b), suggesting this is a clear improvement of CAMSRA. CAMSRA shares the lower sea salt of CIRA in the SH, but is closer to the higher values of MACCRA in the NH. The larger role played by organic matter in CAMSRA reflects the inclusion of a proxy for

anthropogenic SOA added to organic matter, which was missing from the earlier reanalyses. Timeseries and correlation of the Ångström exponent indicate partially better tracing of aerosol size and implicitly of composition variability of CAMSRA than in both CIRA and MACCRA. Validation of AOD with Aeronet data shows there are some hot spots around outgassing volcanoes (in particular Mauna Loa and Mexico City) with high analysis AOD values in CAMSRA that degrade the global



average bias. This is a side effect of possibly erroneous model treatment of diffuse volcanic emissions, i.e. the model-resolution orography not resolving the height of the volcanoes and therefore not being representative of the measurement sites with respect to the volcanic plumes. When calculating global mean statistics, it is advisable to exclude these two stations as unrepresentative.

In addition to being a dataset of better quality and better temporal consistency, CAMSRA has the advantage that it provides more chemical species than CIRA (where only a limited subset was archived) and that data are available at a higher temporal and spatial resolution. In total 56 tropospheric chemical species of the CB05 chemical mechanism, 12 aerosol components and many additional diagnostics such as total columns and extinction coefficients can be obtained from CAMSRA. An inventory

of the available model fields can be found at http://apps.ecmwf.int/data-catalogues/cams-reanalysis/?class=mc&expver=eac4. Users who previously used the MACCRA or CIRA data should note that because of the differences seen between CAMSRA and the older reanalyses, it is not advisable to concatenate data from the older reanalyses with CAMSRA data from more recent years, but that CAMSRA data should be used for the whole period of interest.

The CAMSRA data are freely available (http://atmosphere.copernicus.eu/copernicus-releases-new-global-reanalysis-data-set-atmospheric-composition) and can serve a multitude of users from application developers to scientists and policy makers. The data can be used to analyse the state of the atmosphere or to look at trends that have developed over the past years or decades. Furthermore, the CAMS reanalysis can be used to compute climatologies, evaluate models, benchmark other reanalyses or serve as boundary conditions for regional models for past periods.

One limitation of CAMSRA is that it does not use a stratospheric chemistry scheme (apart form a Cariolle-type parametrization for stratospheric ozone) and the stratospheric concentrations apart from ozone need to be considered with caution. For any future reanalysis, we plan to implement a full stratospheric chemistry scheme and to increase the vertical resolution to bring it in line with the vertical resolution used in ECMWF's NWP system (137 levels, model top at 0.01 hPa). It might also be

beneficial to include the chemistry in the adjoint and tangent linear model of the IFS and to re-calculate the background error statistics for the atmospheric composition variables with the latest version of the model. More time should be spent on acquiring and assessing new observations so that problems like the OMI row anomaly are addressed properly and the quality of the reanalysis is not degraded at times when lower quality data are assimilated (e.g. degraded $NO_2$ analysis during 2003 because of worse quality SCIAMACHY data). It would also be advisable to explore thoroughly the use of reprocessed datasets, e.g.

datasets processed by ESA's CCI and the Seventh Framework Programme (FP7) Quality Assurance for Essential Climate Variables (QA4ECV) project. It could also be investigated if enough atmospheric composition data sets are available prior to 2003 to start a future reanalysis before 2003. Furthermore, work has started to look at emission inversion with the CAMS system and we hope the next reanalysis will include some inversion capability to update the emissions during the assimilation according to the satellite observations.

**Author contributions**

A. Inness prepared the data assimilation of the chemistry fields, ran the reanalysis experiments, carried out a lot of the validation and wrote the manuscript. M. Ades contributed to the assimilation of aerosol data and produced the Aeronet, GAW and WOUDC validation plots, A. Agusti-Panareda and S. Massart provided the code for the TCCON validation, J. Barré and

S. Massart helped with the assimilation of the reactive gases and greenhouse gases, A. Benedictow and M. Schulz provided Fig. 23 and the corresponding text, A.M. Blechschmidt provided Fig. 17, J. Dominguez, M. Razinger and M. Suttie helped





with the acquisition of the data that were assimilated in the CAMS reanalysis, re-archiving of the CAMS reanalysis experiments and data provision to users. R. Engelen and V.-H. Peuch provided useful feedback on the manuscript and help in the coordination of the reanalysis, H.J. Eskes coordinated the CAMS_84 validation activities and provided feedback on the manuscript, J. Flemming and V. Huijnen worked on the chemistry scheme of the IFS and J. Flemming provided Figures 2,
5  6,11,16 &19, L. Jones provided the software for the validation with ozone sondes, GAW, WOUDC, Aeronet and IAGOS data, Z. Kipling and S. Remy worked on the aerosol model and aerosol validation, M. Parrington worked on the emissions and provided Fig.3.

**Acknowledgements**

Thanks to the data providers of the data assimilated in the CAMS reanalysis and the data used for the validation studies in this
10  paper. The Copernicus Atmosphere Monitoring Service is operated by the European Centre for Medium-Range Weather Forecasts on behalf of the European Commission as part of the Copernicus programme (http://copernicus.eu).

**Appendix**

**1 List of GEMS/MACC/CAMS real-time and reanalysis experiments**

| Period | EXP | CLASS | IFS CYCLE | Resolution | Model |
|---|---|---|---|---|---|
| 20080706-20090901 | f1kd | RD | 32R3 | T159/L60 | IFS/MOZART 3.5 coupled system |
| 20090901-20120705 | f93i | RD | CY36R1 | T159/L60 | IFS/MOZART 3.5 coupled system |
| 20120705-20131007 | fnyp | RD | CY37R3 | T255/L60 | IFS/MOZART 3.5 coupled system |
| 20131007-20140224 | fnyp | RD | CY38R2 | T255/L60 | IFS/MOZART 3.5 coupled system |
| 20140224-20140918 | fnyp | RD | CY40R1 | T255/L60 | IFS/MOZART 3.5 coupled system |
| 20140918-20150903 | g4e2 | RD | CY40R2 | T255/L60 | IFS(CB05) |
| 20150903-20160621 | 0001 | MC | CY41R1 | T255/L60 | IFS(CB05) |
| 20160621-20170124 | 0001 | MC | CY41R1 | T511/L60 | IFS(CB05) |
| 20170124-20170926 | 0001 | MC | CY43R1 | T511/L60 | IFS(CB05) |
| 20170926- | 0001 | MC | CY43R3 | T511/L60 | IFS(CB05) |

15  **Table A1: List of GEMS/MACC/CAMS model versions showing the time evolution of the real-time CAMS system since July 2008.**

| Period | Name | EXP | CLASS | IFS CYCLE | Resolution | Model | Production Period |
|---|---|---|---|---|---|---|---|





| 20030101-20090524 | GEMS reanalysis | eac1 | MC | 32R3 | T159/L60 | IFS/MOZART 3.5 coupled system | March 2008 – September 2009 |
|---|---|---|---|---|---|---|---|
| 20030101-20121231 | MACC reanalysis | rean | MC | CY36R1 | T159/L60 | IFS/MOZART 3.5 coupled system | March 2010 – February 2012 |
| 20030101-NRT | CAMS interim reanalysis | eac3 | MC | CY40R2/41R1 | T159/L60 | IFS(CB05) | December 2014 - December 2016, then continued in NRT |
| 20030101-NRT | CAMS reanalysis | eac4 | MC | CY42R1 | T255/L60 | IFS(CB05) | January 2017 onwards |

**Table A2: Reanalyses of atmospheric composition produced with the GEMS/MACC/CAMS system.**

**2 Formulae for calculation of Figure of Merit in space score and Modified normalised mean bias**

The Figure of Merit in space (FMS; Chang and Hanna, 2004) score compares the fit of the model ozone profiles to observation
5  profiles (e.g. ozone sondes) given in partial pressure (milli Pascal), has a score between 1 (perfect fit) and 0 and is defined as

$$CAF = \frac{\int_{\ln(p_{bot})}^{\ln(p_{top})} \min(M, O)}{\int_{\ln(p_{bot})}^{\ln(p_{top})} \max(M, O)} \qquad (1)$$

where M is the model profile, O the observation profile and $p_{top}$ and $p_{bot}$ the top and bottom pressure values of the layer
considered. For Fig. 1 we used $p_{top}$ = 3 hPa and $p_{bot}$ = 1000 hPa.

The Modified Normalized Mean Bias (MNMB) is defined as

$$MNMB = \frac{2}{N} \sum_{i=1}^{N} \frac{m_i - o_i}{m_i + o_i} \qquad (2)$$

with N the number of observations, m the model and o the observed values.

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

.



| | MACCRA | CIRA | CAMSRA |
|---|---|---|---|
| Period covered | 2003 - 2012 | 2003 - present | 2003 - 2016 (will be extended) |
| Assimilation system | IFS Cycle 36r1 4D-Var | IFS Cycle 40r2 (2003-2015) 4D-Var IFS Cycle 41r1 (2016 - …) 4D-Var | IFS Cycle 42r1 4D-Var |
| Horizontal resolution | 80 km globally (T255) | 110 km globally (T159) | 80 km globally (T255) |
| Temporal resolution (Output frequency) | 6-hourly analysis fields 3-hourly forecast fields from 0 UTC up to 24 hours | 6-hourly analysis fields 3-hourly forecast fields from 6 and 18 UTC up to 12 hours | 3-hourly analysis fields 3-hourly forecast fields from 0 UTC up to 48 hours 1-hourly surface forecast fields from 0 UTC up to 48 hours |
| Anthropogenic missions | Chemistry species: MACCity (trend: ACCMIP + RCP8.5), Aerosols: AEROCOM | MACCity (trend: ACCMIP + RCP8.5) & CO emission upgrade Stein et al. (2014) for chemistry and aerosols | MACCity (trend: ACCMIP + RCP8.5) & CO emission upgrade Stein et al. (2014) |
| Biomass burning emissions | GFED (2003–2008) and GFAS v0 (2009–2012) | GFAS v 1.2 | GFAS v 1.2 |
| Biogenic emissions | Monthly mean VOC emissions for the year 2003 calculated by the MEGAN2.1 model (Guenther et al., 2006) used for the whole period. No interannual variability. | Monthly mean VOC emissions calculated by the MEGAN2.1 model (Guenther et al., 2006) using MERRA reanalysed meteorology (Sindelarova et al., 2014) for the period 2003-2010. For the remaining years 2011–2017 a climatology of the MEGAN-MACC data was used. | Monthly mean VOC emissions calculated by the MEGAN model using MERRA reanalysed meteorology (Sindelarova et al., 2014) for 2003-2016. |
| Chemistry modules | CTM MOZART3 coupled to the IFS (see Flemming et al. 2009) | IFS(CB05) (Flemming et al. 2015) & Cariolle ozone parametrisation in stratosphere CHEM_VER=ver14wd | IFS(CB05) (Flemming et al. 2015, with updates documented in Section 2.1.2) & Cariolle ozone parametrisation in stratosphere CHEM_VER=ver15 |
| Aerosol modules | Mocrette et al. (2009) | Mocrette et al. (2009) plus changes described in Flemming et al. (2017) | Mocrette et al. (2009) with changes documented in Section 2.1.1. |
| Input meteorological observations | ECMWF NWP (stream=DA) | ECMWF NWP (stream=DCDA) | As in ERA5 (2003-2016) |
| Assimilated O$_3$ retrievals | GOME, SCIAMACHY, MIPAS, MLS, OMI, SBUV/2 | GOME, SCIAMACHY, MIPAS, MLS, OMI, GOME-2, SBUV/2 | SCIAMACHY, MIPAS, MLS, OMI, GOME-2, SBUV/2 |
| Assimilated CO retrievals | MOPITT, IASI | MOPITT | MOPITT |
| Assimilated NO$_2$ retrievals | SCIAMACHY | --- | SCIAMACHY, OMI, GOME-2 |
| Aerosol used in radiation scheme | Tegen climatology | Tegen climatology | Interactive active aerosols, i.e. aerosol fields from CAMSRA used in radiation scheme |
| Ozone used in radiation scheme | GEMS climatology | GEMS climatology (2003-2015) MACCRA climatology (2016 - ...) | Interactive ozone, i.e. ozone field from CAMSRA used in radiation scheme |
| Stratospheric chemistry | Yes | No, but Cariolle ozone parametrisation in stratosphere and stratospheric O$_3$ available. | No, but Cariolle ozone parametrisation in stratosphere and stratospheric O$_3$ available. |

Table 1: Important commonalities and differences between CAMSRA, CIRA and MACCRA.





| Parameter/ Product | Instrument/ Satellite | Period | Data provider/version | Blacklist criteria | Averaging kernels used | Reference |
|---|---|---|---|---|---|---|
| O₃/ TC | SCIAMACHY/ Envisat | 20020803-20120408 | ESA, CCI (BIRA) | QR>0 SOE<6 | no | Lerot et al. (2009) |
| O₃/ PROF | MIPAS/ Envisat | 20030127- 20040326 20050127-20120331 | ESA, NRT ESA, CCI (KIT) | QR> 0 | no | Von Clarmann et al. (2003, 2009) |
| O₃/ PROF | MLS/ Aura | 20040803-20161231 | NASA, V4 | QR>0 | no | Schwartz et al. (2015) |
| O₃/ TC | OMI/ Aura | 20041001-20150531 20160101-20161231 | KNMI/NASA, V003 NRT | QR>0 SOE<10 | no | Liu et al. (2010) |
| O₃/ TC | GOME-2/ Metop-A | 20070123-20121231 201301-201612 | ESA, CCI (BIRA) fv0100 ESA, CCI (BIRA) fv0300 | QR>0 SOE<10 | No | Hao et al. (2014) |
| O₃/ TC | GOME-2/ Metop-B | 201301-201612 | ESA, CCI (BIRA) fv0300 | QR>0 SOE<10 | no | Hao et al. (2014) |
| O₃/ PC 13L | SBUV/2/ NOAA-14 | 200407-200609 | NASA, v8.6 | QR>0 SOE<6 (SOE<15 between 200407-200409) MODORO > 1000. & PRESS_RL > 450. | No | Bhartia et al. (1996), McPeters et al. (2013) |
| O₃/ PC 13L PC 21L | SBUV/2/ NOAA-16 | 200301-200706 20111201-20130708 20130709-20161231 | NASA, v8.6 NASA, v8.6 NRT | QR>0 SOE<6 (SOE<15 between 200404-200409) MODORO > 1000. & PRESS_RL > 450. | No | Bhartia et al. (1996), McPeters et al. (2013) |
| O₃/ PC 13L | SBUV/2/ NOAA-17 | 200301-201108 | NASA, v8.6 | QR>0 SOE<6 MODORO > 1000. & PRESS_RL > 450. | No | Bhartia et al. (1996), McPeters et al. (2013) |
| O₃/ PC 13L | SBUV/2/ NOAA-18 | 200507-201211 | NASA, v8.6 | QR>0 SOE<6 (SOE<15 from 200404-200409) MODORO > 1000. & PRESS_RL > 450. | No | Bhartia et al. (1996), McPeters et al. (2013) |
| O₃/ PC13 L PC 21L | SBUV/2/ NOAA-19 | 200903-20130708 20130709-20161231 | NASA, v8.6 NRT | QR>0 SOE<6 MODORO > 1000. & PRESS_RL > 450. | No | Bhartia et al. (1996); McPeters et al. (2013) |
| CO/ TC | MOPITT/ Terra | 20020101-20161231 | NCAR, V6 (TIR) | LAT>65. LAT< -65 QR>0 Night time data over Greenland | yes | Deeter et al. (2014) |
| NO₂/ TRC | SCIAMACHY/ Envisat | 20030101-20101231 20110101-20120409 | KNMI V1p KNMI V2 | QR>0 SOE<6 LAT>60 LAT< -60 | yes | Boersma et al. (2004); http://www.temis.nl ; Wang et al. (2008) |
| NO₂/ TRC | OMI/ Aura | 20041001-20101231 20110101-20121231 20130101 -20161231 | KNMI, Domino V1.02 KNMI, Domino V2 KNMI NRT Domino V2 | QR>0 SOE<6 LAT>60 LAT< -60 | yes | Boersma et al. (2006) |
| NO₂/ TRC | GOME-2/ Metop-A | 20070418-20161231 | AC SAF, GDP4.8 | QR>0 SAA | Yes | Valks et al. (2011) |
| NO₂/ TRC | GOME-2/ Metop-B | 201301-20161231 | AC SAF, GDP4.8 | QR>0 SAA | yes | Valks et al. (2011) |
| AOD/ TC | AATSR/ Envisat | 20021201-20120331 | ESA, CCI (Swansea) | abs(LAT)> 70 | no | Popp et al. (2016) |
| AOD/ | MODIS/ | 20021001-20151231 | NASA, COl6 | abs(LAT)> 70 | no | Levy et al. (2018) |





| TC | Terra | 20160101-20161231 | NRT | | | |
|----|-------|-------------------|-----|---|---|---|
| AOD/TC | MODIS/ | 20021001-20151231 | NASA, COl6 | abs(LAT)> 70 | no | Levy et al. (2018) |
| | Aqua | 20160101-20161231 | NRT | | | |

**Table 2: Satellite retrievals of atmospheric composition that were assimilated in the CAMS reanalysis. TC: Total column, TRC: Tropospheric column, PROF: profiles, PC: Partial columns, QR= quality flag given by data providers, SOE: Solar elevation, MODORO: Model orography, PRESS_RL= pressure at bottom of layer, LAT: Latitude, SAA: area of the South Atlantic Anomaly.**

| Station | Latitude, longitude | Reference | MACCRA Bias ± stdv | CIRA Bias ± stdv | CAMSRA Bias ± stdv |
|---------|---------------------|-----------|--------------------|------------------|--------------------|
| Ny-Ålesund | 78.9°N,11.9°E | Notholt et al. (2017a) | -8.03 ± 8.03 | 1.47 ± 4.14 | -1.22 ± 3.62 |
| Sodankyla | 67.37°N, 26.63°E | Kivi et al. (2014) | 3.71 ± 7.19 | 0.85 ± 2.66 | -1.37 ± 3.10 |
| Bremen | 53.1°N,8.85°E | Notholt et al. (2017b) | 5.96 ± 4.91 | 0.88 ± 3.30 | -1.77 ± 3.13 |
| Parkfalls | 45.94°N, 90.27°W | Wennberg et al. (2014) | 4.49 ± 5.04 | 2.99 ± 3.02 | -0.54 ± 2.95 |
| Izaña | 28.3°N,16.5°W | Blumenstock et al. (2014) | 12.92 ± 4.61 | 8.33 ± 3.19 | 4.48 ± 3.27 |
| Darwin | 12.45°S,130.89°E | Griffith et al. (2014) | 4.55 ± 4.50 | 7.30 ± 4.42 | 1.96 ± 5.06 |
| Lauder | 45.04°S,169.68°E | Sherlock et al. (2014) | 4.29 ± 2.67 | 5.47 ± 1.75 | 0.58 ± 1.59 |
| | | | 4.51 ± 2.90 | 3.86 ± 3.04 | 0.71 ± 2.66 |

**Table 3: TCCON stations used in this paper and mean biases and standard deviations from MACCRA, CIRA and CAMSRA in ppb.**

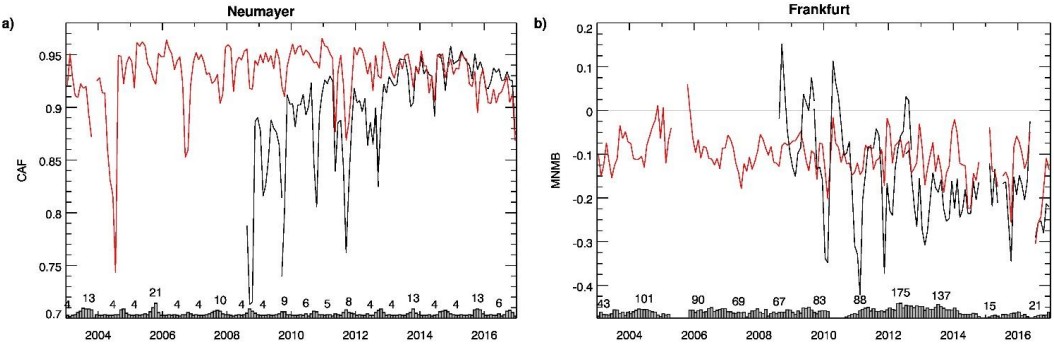

**Figure 1: Timeseries from 2003 to 2016 of (a) FMS score of ozone at Neumayer (1000-3 hPa) against ozone sondes (see section 4.1) and (b) MNMB of CO in the lower troposphere (1000-700 hPa) at Frankfurt airport against IAGOS aircraft data (see section 4.2) from the real-time CAMS system (black) and CAMSRA (red).**




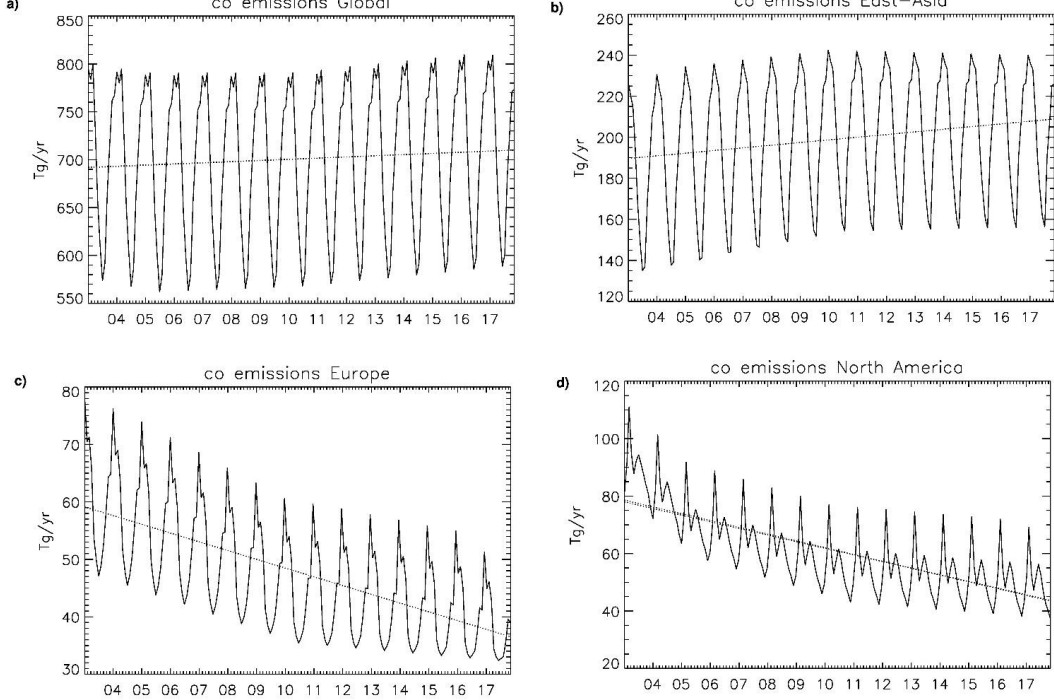

**Figure 2: Monthly CO emissions in Tg/year from anthropogenic sources (MACCITY with correction from Stein et al., 2014) for (a) the Globe, (b) East-Asia, (c) Europe and (d) North America for the period 2003-2016.**





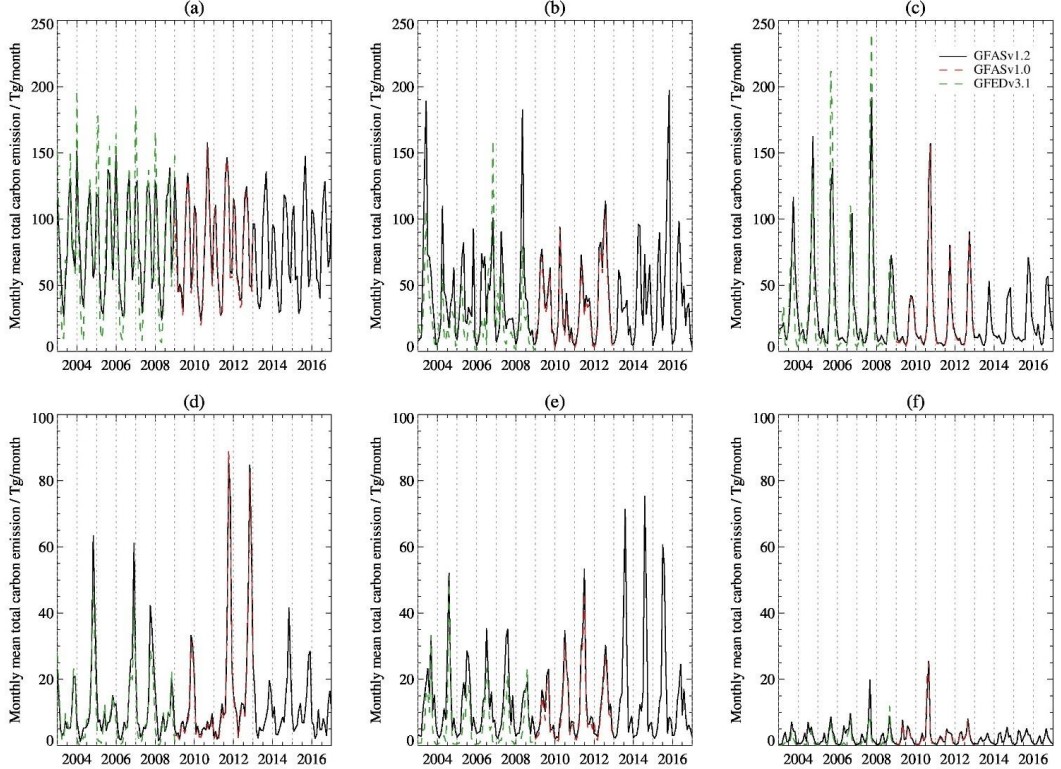

**Figure 3: Time series of monthly total carbon wildfire emissions in Tg/month from GFASv1.2 (2003-2016, black solid line), GFASv1.0 (2009-2012, red dashed line) and GFEDv3.1 (2003-2008, green dashed line) for geographical domains covering: (a) Africa, (b) Asia, (c) South America, (d) Australia, (e) North America, and (f) Europe.**

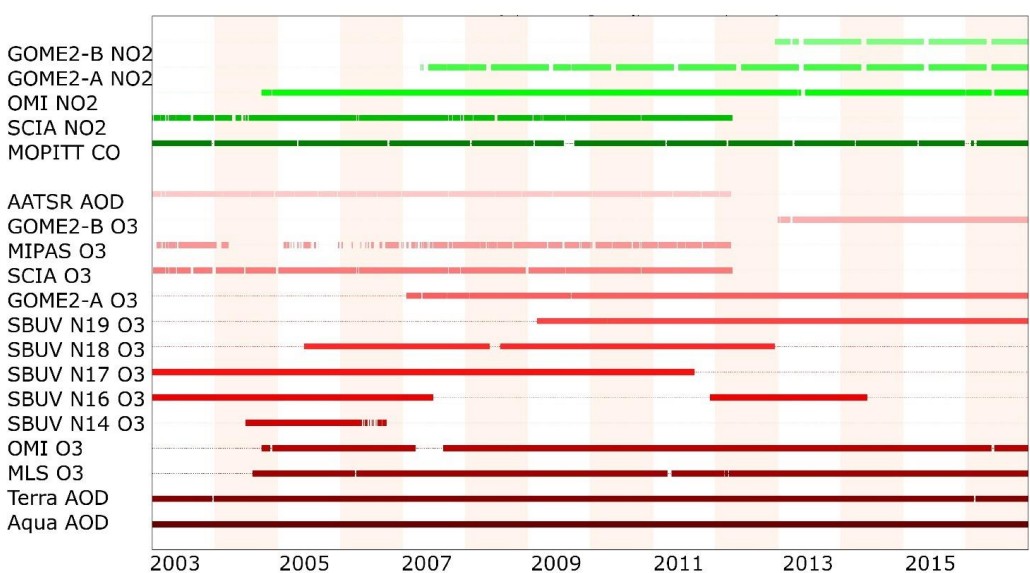

**Figure 4: AC data assimilated in the CAMS reanalysis between 2003 and 2016. In red are shown retrievals for which no averaging kernels were used, in green those where averaging kernels were use.**



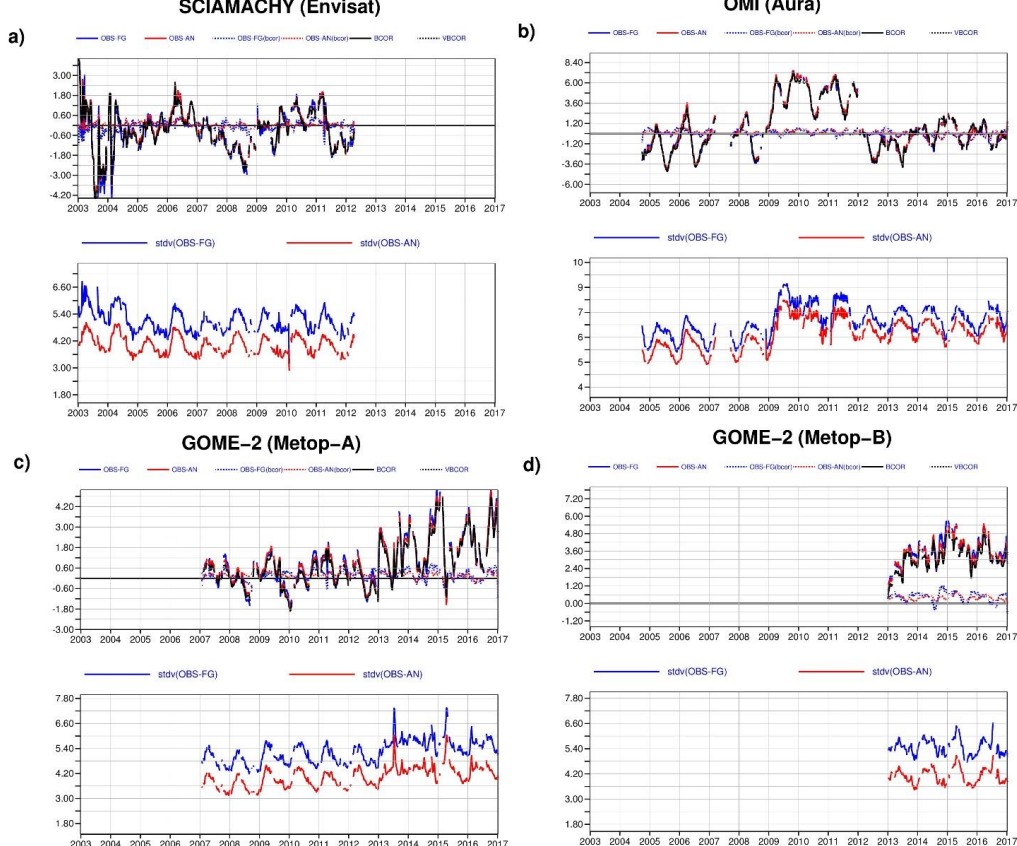

Figure 5: Timeseries of global mean monthly mean TCO3 departures (top plots) and standard deviations of departures (bottom plots) of (a) SCIMACHY, (b) OMI, (c) GOME-2A and (d) GOME-2B. The red lines show analysis departures, the blue lines first-guess departures, black lines bias correction and dotted red and blue lines the bias corrected analysis and first-guess departures, respectively. Values are in DU.





**Figure 6: Seasonally averaged TCO3 from CAMSRA (left), difference between CAMSRA and CIRA (middle) and difference between CAMSRA and MACCRA (right, 2003-2012 only) in DU for the seasons DJF (row 1), MAM (row 2), JJA (row 3) and SON (row 4).**





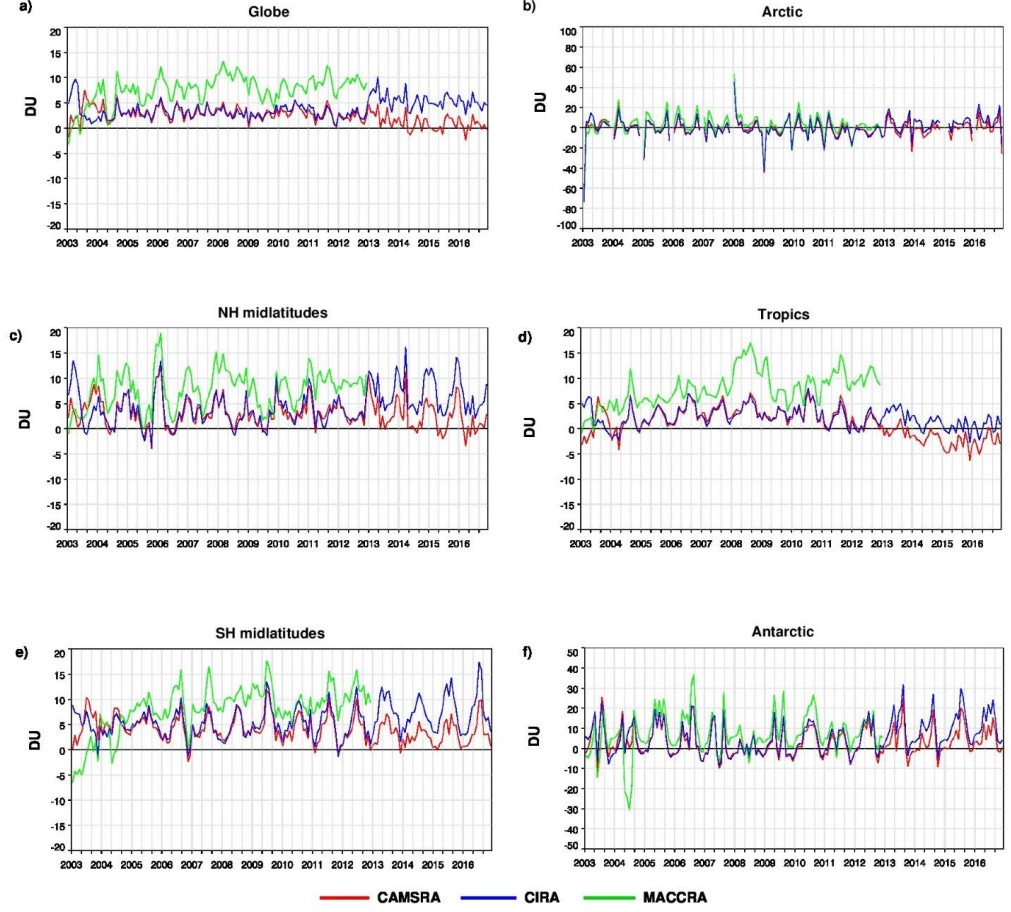

Figure 7: Timeseries of monthly mean TCO3 bias in DU from the three reanalyses compared to WOUDC Dobson data for the areas (a) Globe, (b) Arctic, (c) NH midlatitude, (d) Tropics, (e) SH and (f) Antarctic in DU. About 50-60 stations were available from 2003 to 2014, dropping to about 40 stations after 2014. CAMSRA is shown in red, CIRA in blue and MACCRA in red.





**Figure 8: Mean relative O₃ bias in % between CAMSRA (red), CIRA (blue), MACCRA (green) and ozone sondes averaged over (a) the Globe, (b) Arctic, (c) NH midlatitudes, (d) Tropics, (e) SH midlatitudes and (f) Antarctic. The shaded areas show the standard deviations. For CAMSRA and CIRA the average is calculated over the period 2003-2016, for MACCRA only for 2003-2012.**





**Figure 9: Timeseries of the modified normalized mean bias (MNMB) in the free troposphere (750-300 hPa) of the reanalyses versus ozone sondes for (a) global mean, (b) Arctic, (c) NH midlatitudes, (d) Tropics, (e) SH midlatitudes and (f) Antarctica. CAMSRA is in red, CIRA in blue and MACCRA in green.**





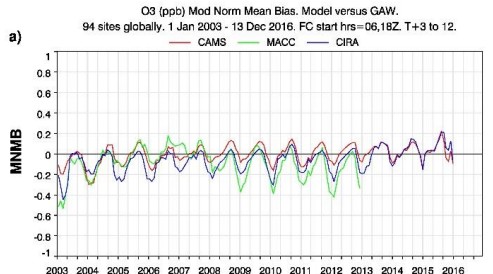
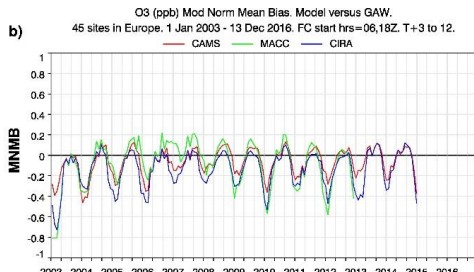

**Figure 10: Timeseries of monthly mean surface ozone MNMB between the three reanalyses and GAW O₃ data averaged over (a) the Globe and (b) Europe. Globally about 60-70 stations were available from 2003 to 2014, dropping to about40-50 in 2015 and then dropping steeply to only a few during 2016. In Europe, the number dropped from 25-35 in 2003-2014 to 17-19 in 2015. CAMSRA is shown in red, CIRA in blue and MACCRA in green.**



**Figure 11: Seasonally averaged TCCO from CAMSRA (left), difference between CAMSRA and CIRA (middle) and difference between CAMSRA and MACCRA (right, 2003-2012 only) in 10$^{18}$ molec/cm$^2$ for the seasons DJF (row 1), MAM (row 2), JJA (row 3) and SON (row 4).**





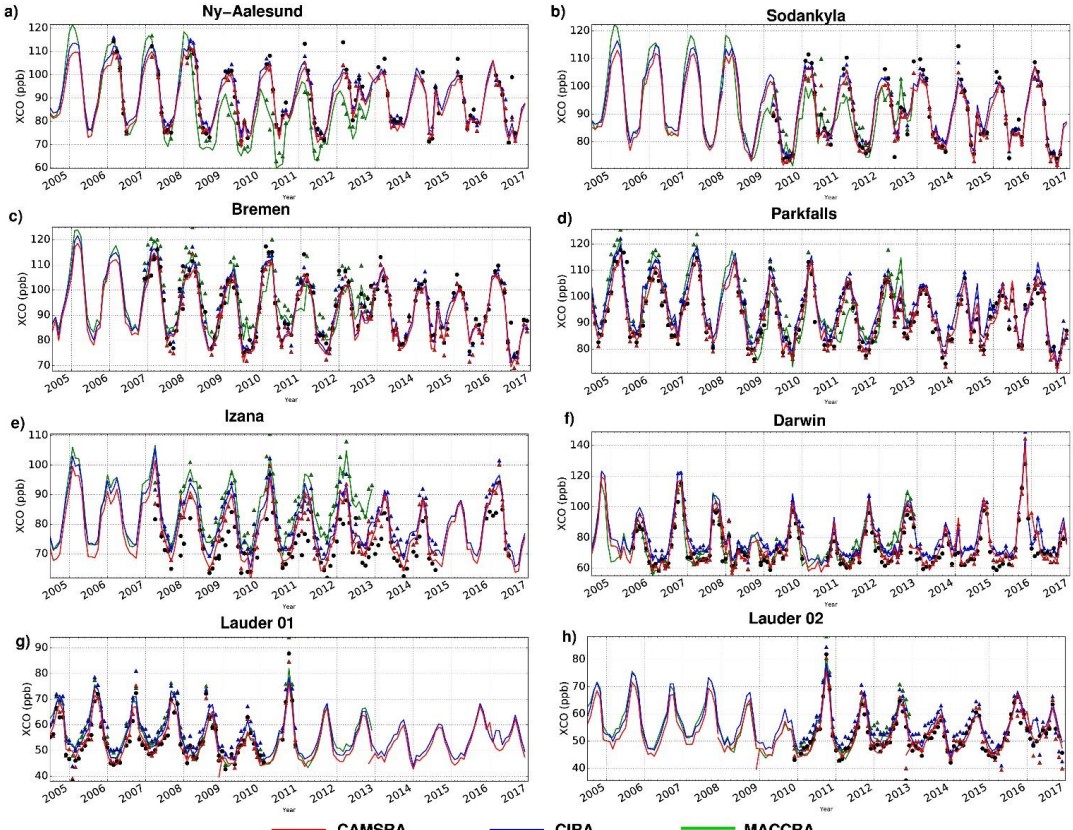

**Figure 12: Column average CO (XCO) in ppb at several TCCON stations. Monthly mean observations are shown by the black dots, corresponding monthly mean XCO columns calculated using the TCCON averaging kernels are shows by the red (CAMSRA), blue (CIRA) and green (MACCRA) triangles. The continuous lines are the monthly XCO for the 3 reanalyses. Show are data for (a) Ny-Ålesund, (b) Sodankyla, (c) Bremen, (d) Parkfalls, (e) Izaña, (f) Darwin, (g) Lauder 2004-2010 and (h) Lauder 2010-2016.**





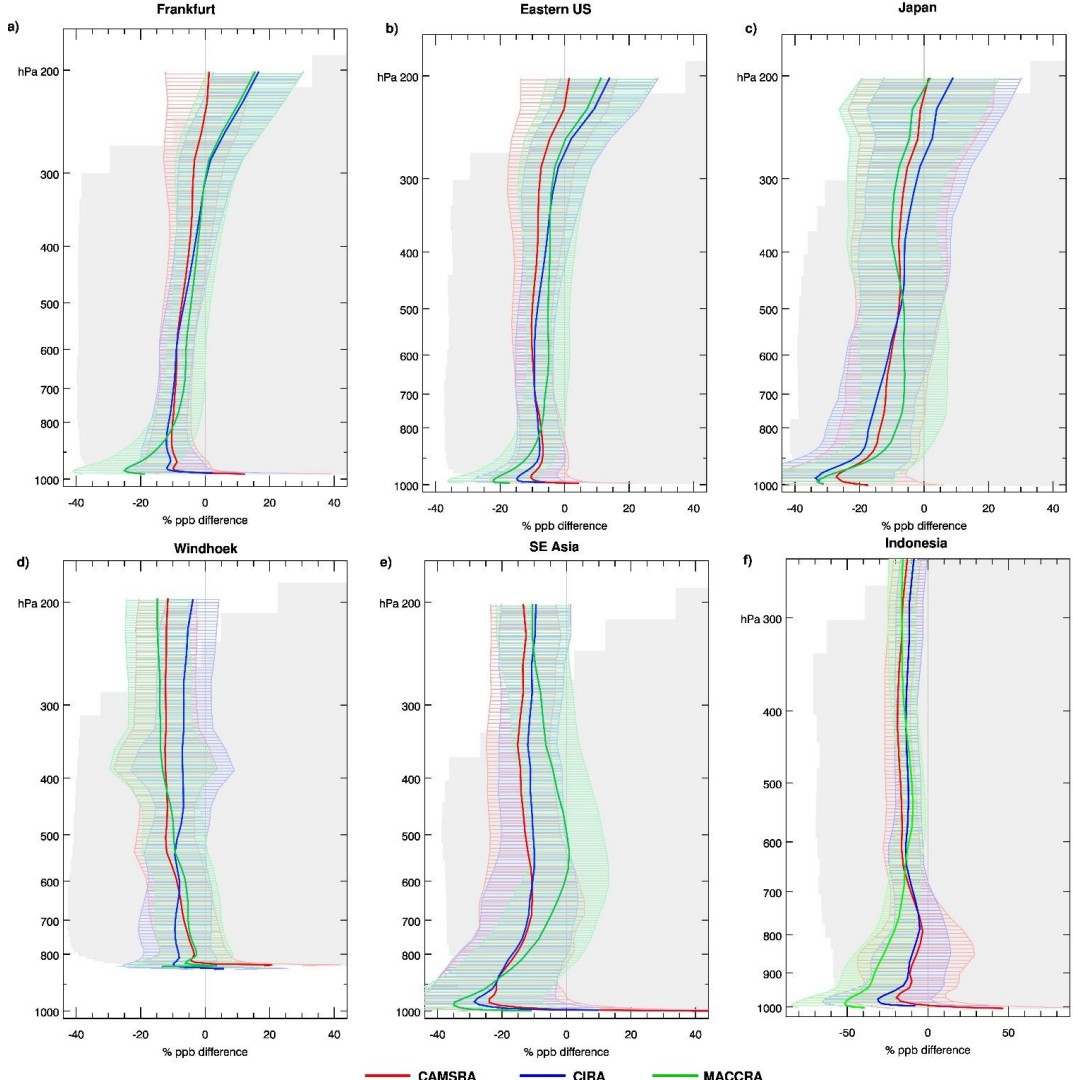

**Figure 13: Mean relative CO bias in % between the reanalyses and IAGOS aircraft data for CAMSRA (red), CIRA (blue) and MACCRA (green) at (a) Frankfurt, (b) Easter US airports, (c) Japanese airports, (d) Windhoek, (e) SE Asian airports and (f) Indonesian airports (note the different scale of the axis for (f)). The shaded areas show the standard deviations. For CAMSRA and CIRA the average is calculated over the period 2003-2016, for MACCRA only for 2003-2012.**



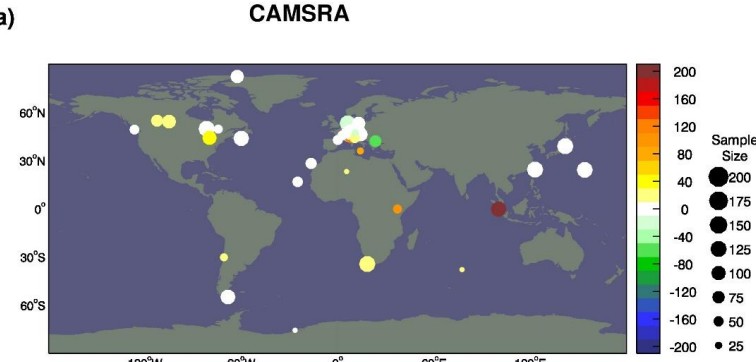

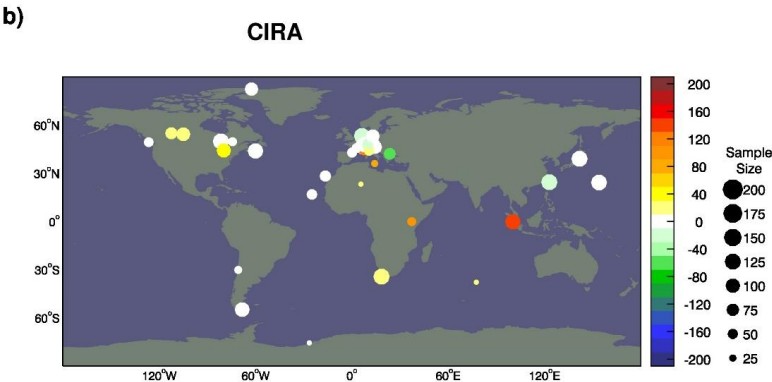

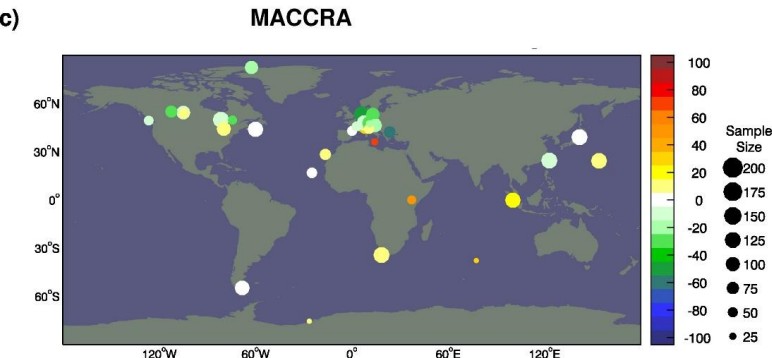

**Figure 14: Mean CO bias in ppb between the three reanalyses and GAW surface observations for (a) CAMSRA, (b) CIRA and (c)**
5   **MACCRA. For CAMSRA and CIRA the average is calculated over the period 2003-2016, for MACCRA only for 2003-2012.**



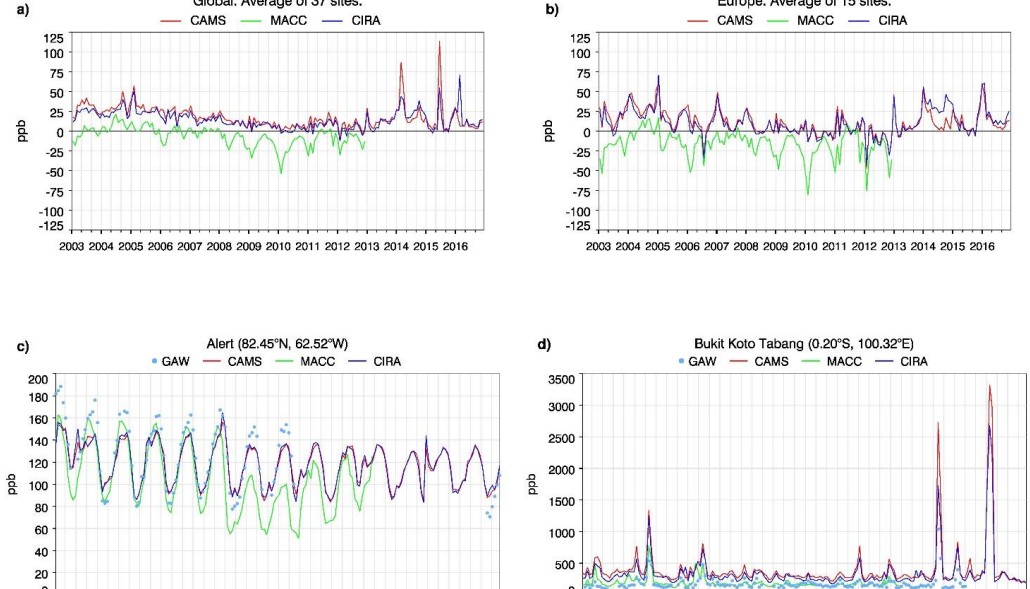

**Figure 15: Top panels: Timeseries of monthly mean surface CO bias in ppb between the three reanalyses and GAW CO data averaged over (a) the Globe and (b) Europe. Between 15-30 stations were available between 2003-2016, with largest number between 2008 and 2014 and smaller numbers in the earlier and later years. Bottom panels: Timeseries of monthly mean CO from GAW (blue dots), CAMSRA, CIRA and MACCRA in ppb at (c) Alert and (d) Bukit Koto Tabang. CAMSRA is shown in red, CIRA in blue and MACCRA in green.**





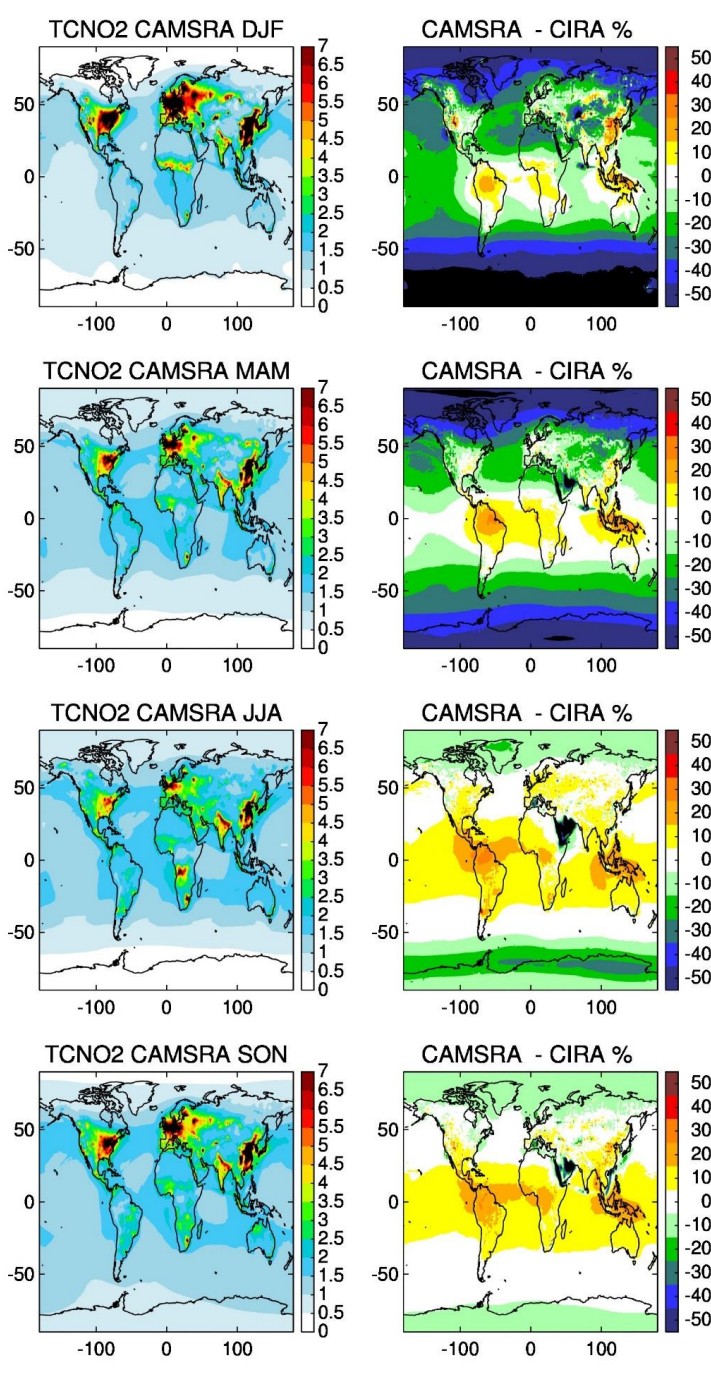

**Figure 16: Seasonally averaged TCNO2 in $10^{15}$ molec/cm$^2$ from CAMSRA (left) and the difference between CAMSRA and CIRA in % (right) for the seasons DJF (row 1), MAM (row 2), JJA (row 3) and SON (row 4) for the period 2003-2016.**





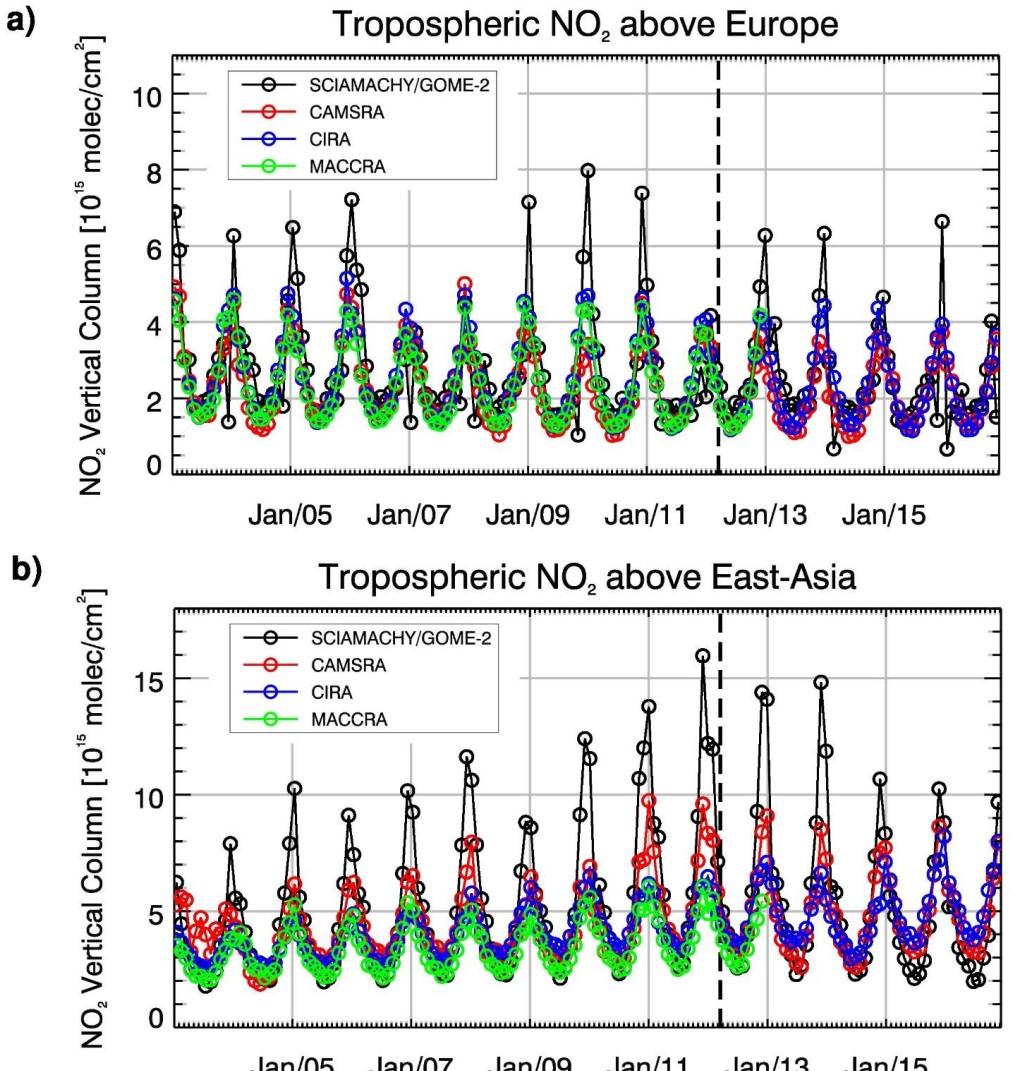

**Figure 17: Timeseries of tropospheric column NO₂ from the three reanalyses and IUB tropospheric NO₂ retrievals in $10^{15}$ molec/cm²
averaged over (a) Europe and (b) East Asia. CAMSRA is shown in red, CIRA in blue, MACCRA in green and the observations in
black.**



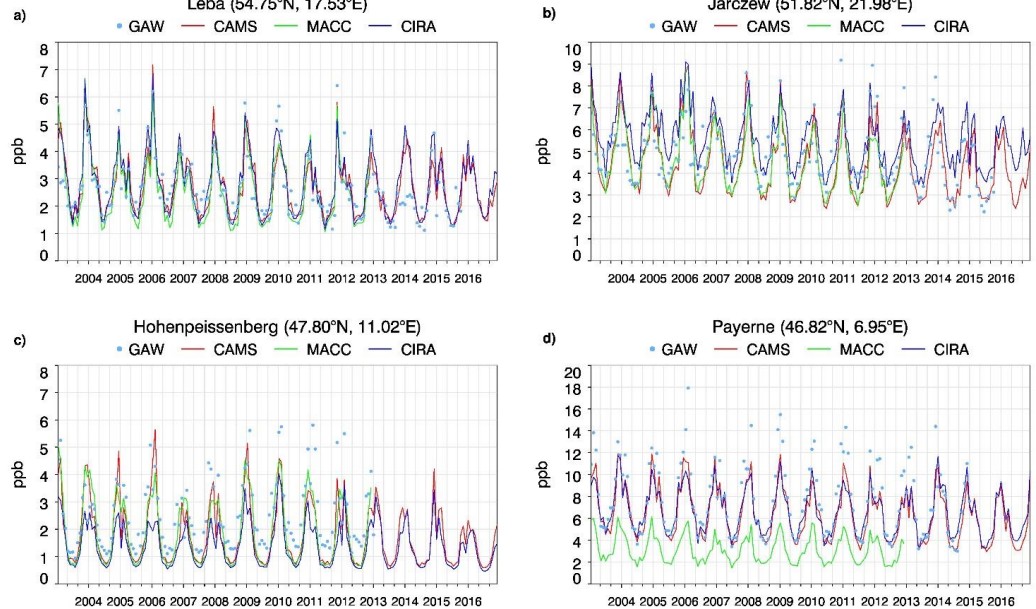

**Figure 18: Timeseries of monthly mean surface NO₂ from CAMSRA (red), CIRA (blue), MACCRA (green) and GAW surface observations (blue dots) for (a) Leba, (b) Jarczew, (c) Hohenpeissenberg and (d) Payerne in ppb. The latitude and longitude of the stations are given in the plot titles.**







**Figure 19: Annually averaged AOD species from CAMSRA (left), difference between CAMSRA and CIRA (middle) and difference between CAMSRA and MACCRA (right, 2003-2012 only) for total AOD (row 1), sea salt (row 2), desert dust (row 3) and sulphates (row 4), organic matter (row 5) and black carbon (row 60). AOD is unitless.**





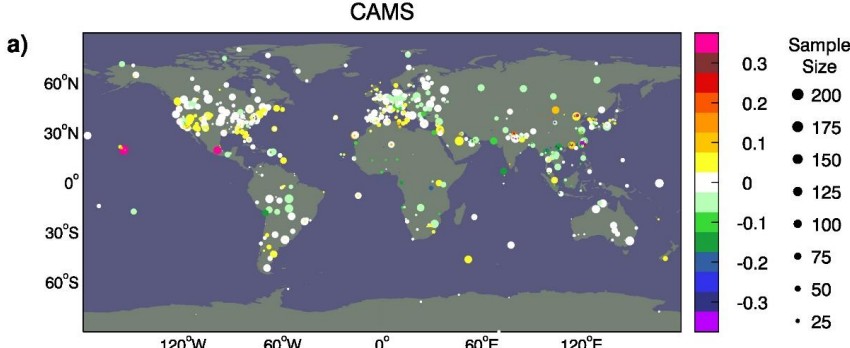

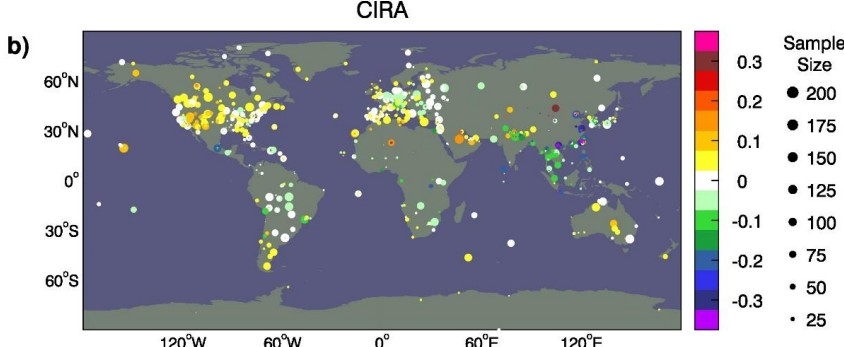

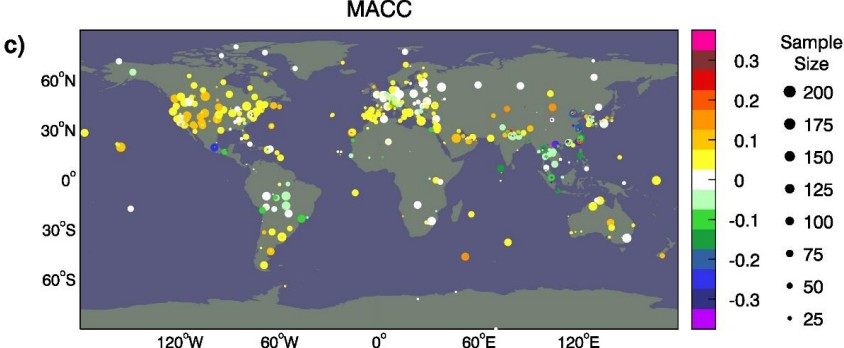

**Figure 20: Mean Total AOD bias between the three reanalyses and Aeronet observations for (a) CAMSRA, (b) CIRA and (c) MACCRA. For CAMSRA and CIRA the average is calculated over the period 2003-2016, for MACCRA only for 2003-2012.**





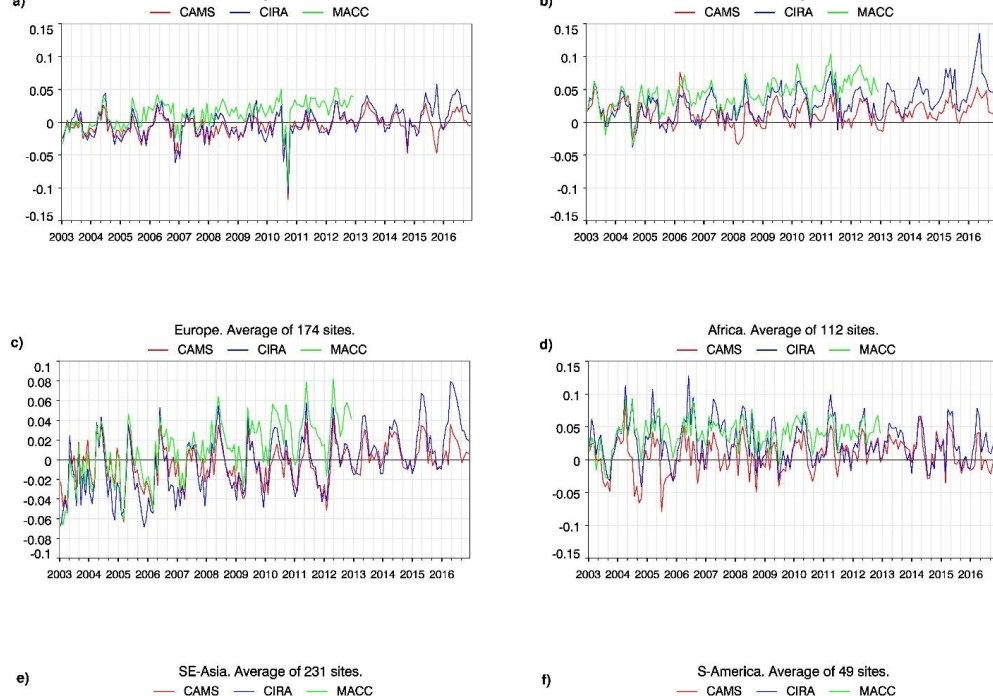

**Figure 21: Timeseries of monthly mean bias of total AOD from the reanalyses against Aeronet observations for the areas: (a) Globe, (b) north America, (c) Europe, (d) SE Asia, (e) Africa and (f) South America. CAMSRA is shown in red, CIRA in blue and MACCRA in green. Mauna Loa and Mexico City were excluded from these timeseries as they are unrepresentative and skew the statistics.**





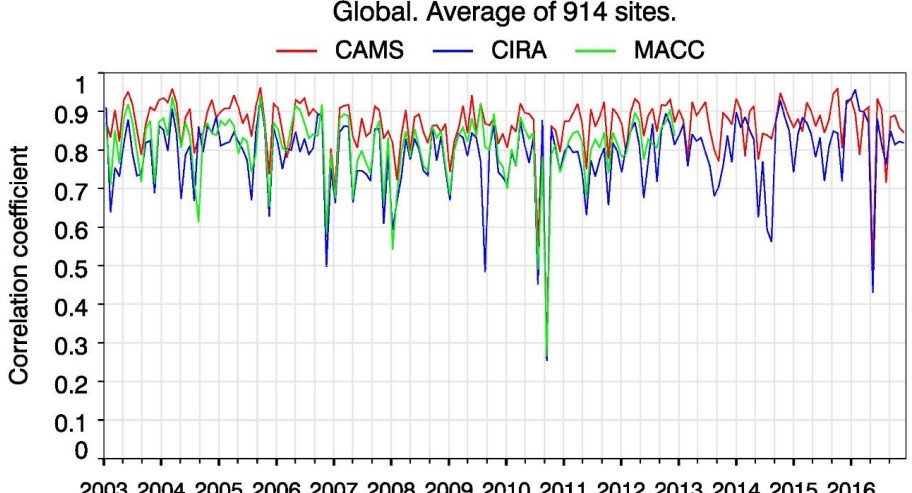

**Figure 22: Timeseries of global correlation coefficient with Aeronet AOD from the three reanalyses. CAMSRA is shown in red, CIRA in blue and MACCRA in green.**

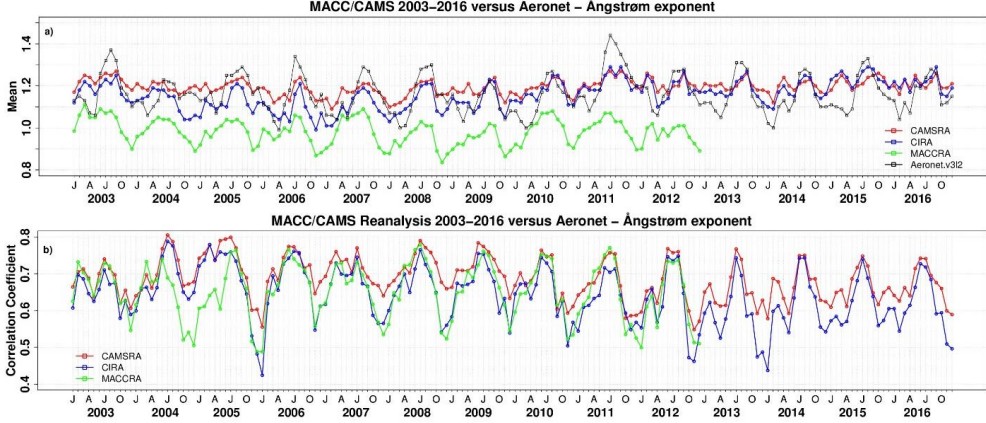

**Figure 23: Evolution of (a) global mean Ångström exponent at Aeronet sites based on matching daily data from model and Aeronet**
10 **and (b) correlation using daily matching Ångström exponent from model and Aeronet (bottom). CAMSRA is shown in red, CIRA in blue, MACCRA in green and Aeronet V3 level 2.0 observations in black.**