# Peer review of "The CAMS reanalysis of atmospheric composition"

_Atmospheric Chemistry and Physics, 2018_

## Short Comment (SC1) · 22 Nov 2018

Great to see an up-to-date atmospheric composition reanalysis from ECMWF with temporally more consistent emissions!

To make the list of existing aerosol reanalyses more inclusive on Page 2 line 35-40, the Navy Aerosol Analysis and Prediction System (NAAPS) aerosol reanalysis product (covering 2003-2015) developed by the US Naval Research Laboratory, is suggested to be included.

Lynch, P., Reid, J. S., Westphal, D. L., Zhang, J., Hogan, T. F., Hyer, E. J., Curtis, C. A., Hegg, D. A., Shi, Y., Campbell, J. R., Rubin, J. I., Sessions, W. R., Turk, F. J., and Walker, A. L.: An 11-year global gridded aerosol optical thickness reanalysis (v1.0) for atmospheric and climate sciences, Geosci. Model Dev., 9, 1489-1522, https://doi.org/10.5194/gmd-9-1489-2016, 2016.

---

## Referee Comment (RC1) · Anonymous Referee #1 · 17 Dec 2018

This work details the new Copernicus Atmosphere Monitoring Service (CAMS) reanalysis of atmospheric composition (CAMSRA) modeling system and compares results to previous atmospheric composition reanalyses from the European Center for Medium-Range Weather Forecasts (ECMWF) (i.e. MACCRA and CIRA). In addition to comparing the three reanalyses products, the paper shows initial validation using independent (non-reanalyzed) observations of ozone, carbon monoxide, nitrogen dioxide, and aerosols (aerosol optical depth). Along the way, the authors caution the reader on how to use the different datasets (e.g. whether or not they can be concatenated, or must be considered completely separately).

This paper will be a useful reference for users of these reanalyses products. I recommend acceptance after a few minor comments are addressed:

* The Table A1 in the Appendix, which details the different model versions, is very

useful.

* Page 2, Lines 32-40: Agree with one comment poster that NRL aerosol reanalysis should be mentioned (Lynch et al., 2016)

* Page 2, Line 36: "also contained aerosols, assimilated concurrently with the meteorology"

* Table 1 greatly helps the reader to understand the differences between the three reanalyses, and is much appreciated.

*Page 7, Line 13: Isn't this still the case in CAMSRA? Or did you not look if sulphates are overstimated?

*Page 8, Line 30: How do you deal with any mis-matches between DT and DB land MODIS observations? Are DB used over DT if there are any coincidences? Similarly, if there are any coincidences between AATSR and MODIS, how do you deal with these?

*Page 9, Line 35: Appreciate putting the monitoring timeseries in the supplement, as it reduces the text nicely

*Page 10, Line 16 - "reanalyse" -> "reanalyses"

*Page 11, Line 31 and Page 12, Line 12 and elsewhere: Appreciate pointing out where user should be aware of issues. If possible, suggest a table highlighting these issues as a useful addition (or a bulleted summary list at the end or in an appendix or supplement).

*Page 16, Line 41: Do you include nitrate aerosol? If not, what biases may this lead to?

*Page 17, Lines 9-20: AERONET is an acronym and should be in all caps
* * *

---

## Referee Comment (RC2) · Anonymous Referee #2 · 6 Jan 2019

The manuscript presents the results from the CAMS reanalysis of atmospheric composition for 2003–2016. The improvements in the reanalysis are evaluated by comparing with the previous reanalysis (MACCRA and CIRA). The CAMS reanalysis provides important information on long-term variations in atmospheric composition. Nevertheless, the evaluation of the reanalysis performance is incomplete and should be improved. More quantitative descriptions are required to summarize the relative performance of the three reanalyses. I would advise the authors to revise the manuscript substantially before considering its publication. My general remarks and specific points are presented below.

1. The descriptions in the text are primarily qualitative. More careful descriptions with summary statistics (e.g., by showing mean bias, RMSE, and temporal correlations for the three reanalyses in tables) are required to make robust conclusions for O3, CO, and NO2, respectively, throughout the manuscript. I would advise the authors to revise

[Figure]

the manuscript substantially by adding more quantitative descriptions and conclusions.

2. It is often summarized that CAMS is better than the two older reanalyses. However, many exceptions exist (e.g., CIRA is better, or CAMS and CIRA are similar), and it is unclear if this can be the general conclusions of this study. The conclusions must be supported by summary statistics for each component of the reanalyses, while more careful discussions and conclusions are required to state how much the reanalysis performance has been improved.

3. A bias correction was applied to the OMI NO2 data using the SCIAMACHY and GOME-2 NO2 data as anchors. Because we expect similar biases between the three NO2 data produced using the same retrieval algorithms, the bias correction could introduce spurious biases into the OMI NO2 data, corresponding to inaccurate diurnal NO2 variations in the model. This could suppress improvements in the afternoon NO2 and photochemical productions of ozone. More careful discussions are required.

Specific comments:

- The discontinuity in the assimilated NO2 products could influence the long-term ozone analysis. This needs to be discussed carefully.

- Section 2.2: It would be useful to show NOx emissions in addition to CO emissions

- Section 2.3: How was the background error covariance (inter-species correlations) constructed? How did the observation information propagate among the species?

- In Fig. 16, it is odd that MACCRA is not shown.

---

## Author Comment (AC1) · 21 Feb 2019

This work details the new Copernicus Atmosphere Monitoring Service (CAMS) reanalysis of atmospheric composition (CAMSRA) modeling system and compares results to previous atmospheric composition reanalyses from the European Center

20 for Medium- Range Weather Forecasts (ECMWF) (i.e. MACCRA and CIRA). In addition to comparing the three reanalyses products, the paper shows initial validation using independent (non-reanalyzed) observations of ozone, carbon monoxide, nitrogen dioxide, and aerosols (aerosol optical depth). Along the way, the authors caution the reader on how to use the different datasets (e.g. whether or not they can be concatenated, or must be

considered completely separately). This paper will be a useful reference for users of these reanalyses products. I recommend

25 acceptance after a few minor comments are addressed:

* The Table A1 in the Appendix, which details the different model versions, is very useful.

* Page 2, Lines 32-40: Agree with one comment poster that NRL aerosol reanalysis should be mentioned (Lynch et al., 2016)

*Done. We have added a sentence: "The US Naval Research Laboratory developed the Navy Aerosol Analysis and Prediction System (NAAPS) aerosol reanalysis product covering the years 2003-2015 (Lynch et al., 2016)."*

30 * Page 2, Line 36: "also contained aerosols, assimilated concurrently with the meteorology"

*We have added that statement.*

* Table 1 greatly helps the reader to understand the differences between the three reanalyses, and is much appreciated.

*Page 7, Line 13: Isn't this still the case in CAMSRA? Or did you not look if sulphates are overstimated?

*Sulphates are reduced in CAMSRA (see Figure 19). We have added a sentence on page 7 at the end of section 2.3: "Sulphates*

35 *are reduced in CAMSRA (see Figure 19 below)."*

*Page 8, Line 30: How do you deal with any mismatches between DT and DB land MODIS observations? Are DB used over DT if there are any coincidences? Similarly, if there are any coincidences between AATSR and MODIS, how do you deal with these?

*We have added this in sections 3.1:*

40 *The data preparation stage for the MODIS observations prioritises Dark Target and will only use Deep Blue data if no Dark Target observations are available. The AATSR and MODIS AOD observations may potentially be coincident but this is dealt*

*with by the data assimilation system. Solving the cost function balances mismatches to both the model and all observations taking in to account both model and individual observation errors.*

*Page 9, Line 35: Appreciate putting the monitoring timeseries in the supplement, as it reduces the text nicely

*Page 10, Line 16 - "reanalyse" -> "reanalyses"

*Changed.*

*Page 11, Line 31 and Page 12, Line 12 and elsewhere: Appreciate pointing out where user should be aware of issues. If possible, suggest a table highlighting these issues as a useful addition (or a bulleted summary list at the end or in an appendix or supplement).

*We have added a bullet list of known issues in the appendix.*

*Page 16, Line 41: Do you include nitrate aerosol? If not, what biases may this lead to?

*Nitrates are not included in the aerosol scheme used in the CAMS reanalysis. They are now being tested in the current model version for implementation in the near-real time CAMS system. We have added this statement in the document:*

*"Nitrates are not yet included in the aerosol scheme. The missing nitrate aerosol is likely to cause an underestimation of total aerosol in the forecast model in regions where nitrate would be a significant component. The total aerosol will be corrected by the assimilation of total AOD observations."*

*Page 17, Lines 9-20: AERONET is an acronym and should be in all caps

*Changed throughout the manuscript.*
The manuscript presents the results from the CAMS reanalysis of atmospheric composition for 2003–2016. The improvements in the reanalysis are evaluated by comparing with the previous reanalysis (MACCRA and CIRA). The CAMS reanalysis provides important information on long-term variations in atmospheric composition. Nevertheless, the evaluation of the reanalysis performance is incomplete and should be improved.

More quantitative descriptions are required to summarize the relative performance of the three reanalyses. I would advise the authors to revise the manuscript substantially before considering its publication. My general remarks and specific points are presented below.

1. The descriptions in the text are primarily qualitative. More careful descriptions with summary statistics (e.g., by showing mean bias, RMSE, and temporal correlations for the three reanalyses in tables) are required to make robust conclusions for O3, CO, and NO2, respectively, throughout the manuscript. I would advise the authors to revise the manuscript substantially by adding more quantitative descriptions and conclusions.

*Our paper is an initial overview and reference paper for the CAMS reanalysis. It is not intended to be a thorough validation paper. Such a paper is being prepared separately and will include a lot more statistics for all the field. There are also a lot more validation plots and statistics in the CAMS validation report (Eskes et al., 2018) that we refer to in our paper. However, we have now calculated summary statistics for several of the Figures (Figures 8, 11,13, 16, 19, 22 in the revised manuscript) and have included tables with the values in the paper.*

2. It is often summarized that CAMS is better than the two older reanalyses. However, many exceptions exist (e.g., CIRA is better, or CAMS and CIRA are similar), and it is unclear if this can be the general conclusions of this study. The conclusions must be supported by summary statistics for each component of the reanalyses, while more careful discussions and conclusions are required to state how much the reanalysis performance has been improved.

*See answer to point 1. We have added tables with summary statistics for several figures.*

3. A bias correction was applied to the OMI NO2 data using the SCIAMACHY and GOME-2 NO2 data as anchors. Because we expect similar biases between the three NO2 data produced using the same retrieval algorithms, the bias correction could

introduce spurious biases into the OMI NO2 data, corresponding to inaccurate diurnal NO2 variations in the model. This could suppress improvements in the afternoon NO2 and photochemical productions of ozone. More careful discussions are required.

*We have not validated the diurnal cycle of NO2 as we do not have observations available that would allow us to do this. Hopefully this will be included in the validation paper that is under preparation. As the assimilation of NO2 usually has a small impact in the CAMS system because of its short life time we do not expect this to be a problem.*

Specific comments:

- The discontinuity in the assimilated NO2 products could influence the long-term ozone analysis. This needs to be discussed carefully.

*As we do not have an additional experiment where we assimilate O3 data, but no NO2 data it is not possible to infer this information from the CAMS reanalysis. We have seen in the past (Inness et al., 2015) that the impact of the NO2 assimilation is generally small, so we do not expect this to lead to problems in the ozone analysis. We agree that it would make a very interesting case study to investigate this by running a series of additional longer experiments without NO2 assimilation, but this is not within the scope of this paper.*

- Section 2.2: It would be useful to show NOx emissions in addition to CO emissions

*We have added a plot with MACCity NO emissions in Section 2.2 and changed the related text to:*

*The anthropogenic MACCity emissions for CO are shown in Fig. 2 and for NO in Fig. 3. The CO emissions decrease over Europe and North America in the range of 1 to 5 % per year, but increase over South East Asia by a similar amount. The global trend for CO is close to zero. The MACCity NO emissions decrease with time over Europe and North America, but increase over East Asia until 2015 which is in contrast to satellite derived emission inventories that show a peak over China in 2012 (e.g. Ding et al., 2017).*

- Section 2.3: How was the background error covariance (inter-species correlations) constructed? How did the observation information propagate among the species?

*There are no inter-species correlations in the CAMS background error covariance matrix. It is univariate. We have added this sentence to section 2.3:*

*The background errors for the chemical species are univariate; i.e. the error covariance matrix between chemical species or between chemical species and dynamical fields is diagonal. Hence each species is assimilated independently from the others.*

- In Fig. 16, it is odd that MACCRA is not shown.

*We have changed this Figure (now Fig. 17) and are now showing tropospheric column NOx from CAMSRA as well as the differences between CAMSRA-CIRA and CAMSRA-MACCRA. We have changed the text accordingly.*

[revised manuscript text omitted]

---

## Author Response (AR2)

We have added the requested changes. They can be seen below as the differences between the previously submitted manuscript and the latest version.

[revised manuscript text omitted]